# Interactive Classification with Real-Time Contrastive Explanations

## Abstract

We propose a framework in which users collaborate with machines to solve classification tasks, aided by contrastive explanations. Among these, counterfactual explanations stand out for their intuitiveness and effectiveness. However, long-standing challenges in counterfactual generation involve the efficiency of the search process, the likelihood of generated instances, their interpretability, and in some cases, the validity of the explanations themselves. In this work we address all these issues to present the first generative framework suited for real time explainable interactive classification. Our method leverages a label disentangled regularized autoencoder to achieve two complementary goals: generating likely instances according to the learned distributions and promoting label discrimination to enable precise control over the decision boundary. By modeling the class-conditional data distribution, the framework avoids computationally expensive gradient-based optimizations, instead directly generating explanations based on the modelled counterfactual class distribution. A user study on a challenging human-machine classification task demonstrates the approach's effectiveness in enhancing human performance, emphasizing the importance of contrastive explanations.

## 1 Introduction

The advances of the past years in machine learning and the field of AI allowed models to improve drastically in the most disparate tasks leading to these models being eventually able to overcome humans' ability and understanding in specific domains (Taigman et al., 2014; He et al., 2015; Esteva et al., 2017; Rajpurkar, 2017; Matek et al., 2019). With this in mind, fostering synergistic collaboration between humans and AI has become a priority to enhance users' ability to tackle critical tasks. However, implementing environments where humans and machine learning models work together to solve problems has proven to be highly challenging. For this reason, Explainable AI arose from the need of transparency and to improve understanding of what are known as black-box models (Gunning et al., 2019). With the goal of explaining the inner workings of deep-learning models, researchers have provided users with many different techniques of post-hoc explanations. Among these, counterfactuals consist of instances describing the necessary changes in input features that alter the prediction to a predefined output (Molnar, 2022), and are especially appealing for a human decision maker (Fernández-Loría et al., 2021). Counterfactual explanations should carry the following properties: i) *validity* – the model prediction on the counterfactual instance needs to follow a predetermined class; ii) *interpretability* – the explanatory instance should be interpretable, iii) *likeliness* – the explanation should be representative of the counterfactual class distribution, iv) *proximity* – the counterfactual instance should be similar to the original one.

Despite the appeal of counterfactual explanations, existing approaches have struggled in satisfying the desired properties, especially likeliness (Poyiadzi et al., 2020; Dhurandhar et al., 2018), actionability (Guidotti et al., 2019; Dhurandhar et al., 2019) or proximity (Guidotti, 2022) of the counterfactual being generated. Efficiency in generation is another major problem of existing solutions (Farid et al., 2023; Wachter et al., 2017; Kanamori et al., 2020) undermining the potential of explanations in real-time interactive settings. Simultaneously, generative models in XAI are gaining attention for improving explanation quality (Schneider, 2024). Inspired by this, we propose a generative framework for interactive classification that leverages

counterfactual explanations that satisfy the mentioned properties and that are computationally efficient, so to allow a real-time collaboration with users.

Our framework builds upon the work of Zheng & Sun (2019) and utilizes a regularized autoencoder with a latent space explicitly disentangled into label-relevant and label-irrelevant dimensions (hereafter referred to as label disentangled for brevity). This allows to learn class-specific representations and enables the generation of counterfactuals by simply trading-off the likelihood of the explanation according to the modelled counterfactual label distribution with its proximity to the instance to explain. *Likeliness* of the output is assured by the underlying generative model, *validity* is guaranteed by the explicit modeling of the decision boundary between classes and *proximity* is additionally encouraged by combining label-relevant latent dimensions with label-irrelevant ones, which are shared among classes. *Efficiency* is achieved by directly generating explanations according to the learned counterfactual label distribution, thus sidestepping expensive gradient based optimizations. Finally, *interpretability* of explanations is improved extracting interpretable concepts associated to the latent dimensions and presenting the most relevant conceptual changes together with the counterfactual image.

To the best of our knowledge, our contribution is the first approach to explainable interactive classification suited for a real-time user interaction. We assess its effectiveness through a user study in which participants tackle a challenging task in collaboration with our support system. The study results clearly demonstrate the potential of our approach in enhancing human performance and highlight the crucial role of counterfactual explanations in achieving these improvements.

## 2 Related work

**Interactive Classification**  Interactive classification aims at improving users performance on classification tasks by providing users feedback from an underlying machine learning model. Many user studies evaluate the effect of model predictions or various explanatory techniques on users but such approaches lack an interactive component in the study configuration (Bansal et al., 2021; Buçinca et al., 2021; Bussone et al., 2015; Das & Chernova, 2020; Feng & Boyd-Graber, 2019; Guo et al., 2019; Kulesza et al., 2012; Lee et al., 2019; Levy et al., 2021; Liu et al., 2021; Park et al., 2019; Alqaraawi et al., 2020; Bansal et al., 2021; Hohman et al., 2018; Chromik et al., 2021; Weerts et al., 2019; Hase & Bansal, 2020; Ribeiro et al., 2018; Buçinca et al., 2020; Cai et al., 2019a;b; Dodge et al., 2019; Kulesza et al., 2013; Lai & Tan, 2019). Other approaches leverage contrastive explanations (De-Arteaga et al., 2020; Lucic et al., 2020; Binns et al., 2018; Ehrlich et al., 2011; Lim et al., 2009; Wang & Yin, 2021; Cohen et al., 2021) but no approach has yet tackled interactive classification for the image domain.

**Contrastive explanations**  Contrastive explanations are among the most widely studied forms of explanation. Prior work (Feghahati et al., 2020; O'Shaughnessy et al., 2020; Samangouei et al., 2018) has demonstrated their effectiveness in enhancing the interpretability of AI models compared to other explanation methods, while Dhuliawala et al. (2023) has shown that human users prefer them over alternative forms of explanation. Motivated by these findings, we propose a framework specifically designed to support a novel and efficient technique for generating counterfactual examples. We introduce related work in the field of counterfactual explanations and clarify what specific limitations of current approaches motivate implementing our approach for the interactive classification setting. More precisely, contrastive explanations aim at justifying a choice by rejecting the other viable options. Throughout the years, various techniques have been proposed to achieve this goal (Prabhushankar et al., 2020; Wang & Wang, 2022; Jacovi et al., 2021; Miller, 2021), with counterfactuals being the most popular option. With the growing use of Deep Generative Models, such as Generative Adversarial Networks (GANs) (Goodfellow et al., 2014) and VAEs (Kingma & Welling, 2013; Rezende et al., 2014), to explain model decisions, the most common approach has been to progressively modify the input to reveal the most meaningful and interpretable changes (Feghahati et al., 2020; Joshi et al., 2019; Liu et al., 2019; O'Shaughnessy et al., 2020; Samangouei et al., 2018; Szegedy et al., 2013). However, these operations can be computationally intensive and often require complex gradient-based optimizations, as seen in Poels & Menkovski (2022) or in Luss et al. (2021), where concepts extracted from a disentangled VAE are central to the explanation process. This computational bottleneck hinders interactive classification by significantly slowing down explanations generation, making real-time interaction impractical.

To address this, we propose an alternative optimization technique that is both efficient and facilitates immediate feedback from the machine. More recent approaches leverage knowledge of causal graphs (Pawlowski et al., 2020; Ribeiro et al., 2023; Dash et al., 2022; Kocaoglu et al., 2017; Kladny et al., 2023) and propose explanatory pipelines that allow direct causal interventions. In our proposal, we relax this requirement since such information is rarely available in real-world datasets. Instead, we introduce a methodology that enhances applicability while maintaining a strong focus on interpretability, linking generated explanations to learned concepts via a concept-relevance metric we present. This last component of our explanatory pipeline is designed to tackle the main issue of works leveraging denoising diffusion probabilistic models (DDPMs) (Ho et al., 2020; Song et al., 2020) for counterfactual explanations (Jeanneret et al., 2022; 2023; Augustin et al., 2022; Farid et al., 2023). More precisely, despite the exceptional performance of DDPMs that allows generation of very realistic counterfactuals, the resulting explanations are not clear regarding which features have been changed and how changes reflect in the target model seriously undermining their interpretability.

**Generative AI and disentanglement** Disentanglement plays a central role in the framework we propose, in terms of both learning disentangled latent representations and disentangling between label-relevant and label-irrelevant dimensions in the latent space. We now present current approaches and limitations of research in this field that can apply to our proposal. Disentangled feature representations, or high level generative factors in disjoint subsets of the feature dimensions, carry many desirable properties such as intervention and interpretability (Kumar et al., 2017; Bengio et al., 2013). An important results comes from Locatello et al. (2019) who show that it is not always possible to construct disentangled embedding spaces as the problem is inherently unidentifiable without additional assumptions such as observed variables (Hyvärinen & Pajunen, 1999; Kazhdan et al., 2020) or tuples of observations that differ in only a limited number of components (Locatello et al., 2020). Leemann et al. (2023) argue that concept discovery should be identifiable and propose two provably identifiable concept discovery methods for components that are not correlated or do not follow a Gaussian distribution. Unsupervised approaches that leverage VAEs (Higgins et al., 2017; Kumar et al., 2017; Chen et al., 2018; Kim & Mnih, 2018) instead incorporate additional regularization components or derive alternative ELBO formulations. Not surprisingly, a body of works exploiting classification losses to encourage a disentangled latent representations at a label level already exists (Dhuliawala et al., 2023; Ding et al., 2020; Zheng & Sun, 2019). However, the two contributions of Dhuliawala et al. (2023) and Ding et al. (2020) are conceived for classification and cannot generate new instances, while the one of Zheng & Sun (2019) can perform generation but is designed to optimize quality of generated images exploiting high-dimensional latent spaces, making it unsuited for interpretable concept extraction.

**Deterministic regularized autoencoders** Deterministic regularized autoencoders (RAE) were first introduced by Ghosh et al. (2019) as alternative decoder regularization schemes with respect to the original noise injection mechanism first proposed in the VAE formulation. Such models require an additional density estimation step to be able to sample latent codes to be reconstructed. Alternative more complex unsupervised approaches (Saseendran et al., 2021; Böhm & Seljak, 2020; Ghose et al., 2020) have been proposed over the years to side-step ex-post density estimation by shaping the latent space according to a uni-modal or multi-modal distribution. Being unsupervised, these approaches do not allow to perform disentanglement at a label level, which is essential for counterfactual explanations. Our approach builds on these ideas and adapts them to the supervised setting.

## 3 Method overview

In this section we present an overview of the methodology we propose. More precisely, our framework is centered around a label disentangled RAE that simultaneously learns a generative process and a classification task. This allows class distributions to guide both the label predictions and their explanatory process. (For simplicity, we will refer to this framework as the disentangled RAE moving forward). On the other hand, the goal of the explanatory component of the framework is to answer contrastive questions such as: *Why $P$ rather than $Q$?* Where $P$ is a *fact* and $Q$ is an hypothetical alternative, or *foil* (Miller, 2021). In the context of interactive multi-label classification, $P$ corresponds to the machine's prediction, and $Q$ represents a user-choice. To provide an explanation, we generate a counterexample that the model would classify as $Q$, highlighting the contrast with $P$. We show that this can be efficiently achieved to allow real-time interaction

with a novel counterfactual generation technique that operates under the assumption that data follows a mixture of Gaussian distributions—an assumption explicitly supported and enforced by our approach. The corresponding counterfactual search process thus consists in three steps: i) identification of a set of candidate counterfactuals according to the criteria of *proximity* and *likeliness*; ii) extraction of the expected value of the set under the alternative class distribution as the generated counterfactual; iii) computation of the top-$k$ most impactful changes in the latent space as interpretable concept changes explaining the counterfactual. This framework aims at capitalizing on the following advantages:

- *Proximity*: Our method optimizes the trade-off between *likeliness* and *proximity* in the latent space. Additionally, explanations share part of their latent representation with the original instance, ensuring a natural connection between the two;

- *Interpretability*: Extracting interpretable concepts via latent traversal allows to provide an intelligible feedback to users in terms of relevant components of the visual counterfactual explanation;

- *Validity*: the assumptions of the predictive model are coherent with the ones of the chosen explanatory technique, allowing full control over the predictive mechanism;

- *Likeliness*: learning the latent-space data distribution allows for fast, efficient and likely counterfactuals generation with the methodology we propose.

The full interactive classification pipeline, shown in Figure 1(a), also displays the explanatory process, which can be divided in three main steps: an encoding step, a counterfactual search step and a decoding step. In the following, we describe the generative model and the training methodology we employ, we present the novel counterfactual generating technique and illustrate the findings of the user study we conducted.

## 4 Denoising Disentangled Regularized Autoencoders

The generative model in our explanatory pipeline consists of a disentangled regularized autoencoder. Our architecture, shown in Figure 1(b), includes a label-relevant encoder $\text{ENC}_s(\cdot)$, that leverages label supervision to map inputs to a latent representation that follows a mixture of Gaussians. Additionally, the architecture features a label-irrelevant encoder $\text{ENC}_u(\cdot)$, which uses adversarial classification to learn high-level generative factors shared across labels. Training occurs in two stages. First, label-relevant and label-irrelevant dimensions are jointly used for reconstruction by the decoder $\text{DEC}(\cdot)$. We refer to this intermediate model as deterministic disentangled autoencoder, as it is not suited for generation. In the second stage, we extract latent representations and employ a noise injection mechanism to create a smooth latent space. We leverage the auxiliary model to handle the noise and achieve decoder regularization by reconstructing denoised representations. We now introduce the necessary background, and then present the deterministic and generative training procedures.

### 4.1 Background

VAEs are a type of parametric model following an encoding $q_\phi(z|x)$ and decoding $p_\theta(x|z)$ mechanism, trained with the goal of maximizing likelihood of evidence through its lower bound (ELBO):

$$\log p(x) \geq \mathbb{E}_{q_\phi(z|x)}[\log p_\theta(x|z)] - D_{\text{kl}}(q_\phi(z|x) \parallel p(z)) \tag{1}$$

where $\phi$ and $\theta$ are the parameters of the encoder and decoder respectively. According to such formulation, $\mathbb{E}_{q_\phi(z|x)}[\log p_\theta(x|z)]$ is the reconstruction loss ($\mathcal{L}_{\text{REC}}$), which encourages encoded inputs to be decoded with fidelity, and $D_{\text{KL}}(q_\phi(z|x) \parallel p(z))$ is the Kullback-Leibler divergence between the output of the recognition model $q_\phi(z|x)$ and the prior latent distribution $p(z)$. The former is extracted from the encoder, which returns mean $\mu_\phi(x)$ and variance $\Sigma_\phi(x)$ parameters through which the latent code $z$ is sampled for every input $x$, while the latter is typically modelled as a standard Gaussian.

The ELBO objective can be extended to incorporating classification terms as in Zheng & Sun (2019), with the idea of disentangling the latent space via label supervision. A common choice is to exploit the Gaussian

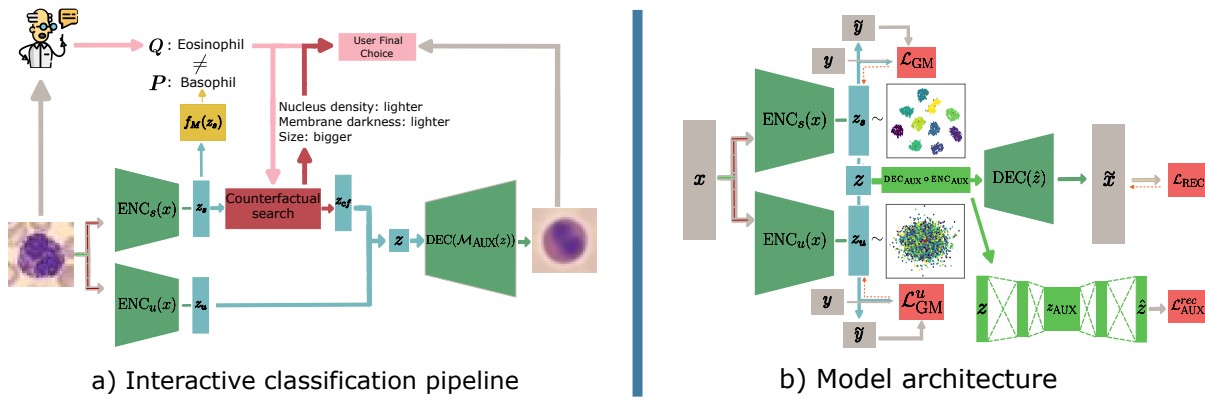

Figure 1: a) Our interactive explainable classification pipeline, consisting of the encoding, counterfactual search and decoding steps; b) Denoising disentangled regularized autoencoder architecture.

mixture framework of Wan et al. (2018) who propose to apply an alternative loss $\mathcal{L}_{\text{GM}}$ to the latent representation $z_i$ of instance $x_i$ with label $y_i$. The first component of the loss is a Gaussian classification term and a the second one is a likelihood regularization term responsible of efficiently shaping the latent space according to a mixture of Gaussian distributions:

$$\mathcal{L}_{\text{GM}} = -\frac{1}{N} \sum_c \sum_i \mathbb{I}(y_i = c) \, \log \frac{\mathcal{N}(z_i; \mu_{y_i}, I) p(y_i)}{\sum_c \mathcal{N}(z_i; \mu_c, I) p(c)} + N \, \log \mathcal{N}(z_i; \mu_{y_i}, I) \tag{2}$$

where the mean $\mu_c$ parameters are encoding statistics accumulated during training while assuming identity covariance matrices.

## 4.2 Training Deterministic Disentangled Autoencoders

The first stage of training combines reconstruction, classification, and regularization objectives to efficiently shape the label-specific latent space as a mixture of Gaussians, achieving strong classification performance while encouraging a smooth latent structure. For the label-irrelevant loss, focused on learning high-level representations shared across classes, we apply Gaussian classification to the output of the label-irrelevant encoder within the Gaussian mixture framework. The key difference is that the posterior class probabilities are expected to follow a uniform distribution:

$$\mathcal{L}_{\text{GM}}^u = -\frac{1}{N} \sum_i \sum_c \frac{1}{|\mathcal{C}|} \, \log \frac{\mathcal{N}(z_i; \mu_c, I) p(c)}{\sum_c \mathcal{N}(z_i; \mu_c, I) p(c)} + N \, \log \mathcal{N}(z_i; 0, I) \tag{3}$$

The final loss is defined as follows:

$$\mathcal{L}_{\text{DET}} = \mathcal{L}_{\text{REC}} + \lambda_s \mathcal{L}_{\text{GM}} + \lambda_u \mathcal{L}_{\text{GM}}^u \tag{4}$$

The pseudocode of the training procedure is shown in Algorithm 3 in Appendix C.1. In the following we show how to transition from a deterministic to a generative model.

## 4.3 From Deterministic to Generative Disentangled Autoencoders

The deterministic disentangled autoencoder model is not suited for generation. For this reason, and inspired by highly performing DDPMs (Ho et al., 2020), we propose an alternative approach to latent space smoothing based on denoising autoencoders. We argue that with a single noise injection step it is possible to effectively

transition from a deterministic to generative model. We treat noise as a hyper-parameter and the structure of the already learned latent space significantly simplifies the regularization task. More precisely, we process stochastic representations with an auxiliary model $\mathcal{M}_{\text{AUX}} : \text{DEC}_{\text{AUX}} \circ \text{ENC}_{\text{AUX}}$ and reconstruct denoised latent representations. Given latent dimension $z$, noise $\epsilon \sim \mathcal{N}(0, I)$ and noise parameter $\sigma$ we define:

$$\sigma\hat{\epsilon} = z + \sigma \cdot \epsilon - \text{DEC}_{\text{AUX}}(\text{ENC}_{\text{AUX}}(z + \sigma \cdot \epsilon))$$
$$\mathcal{L}_{\text{AUX}}^{rec} = \sigma^2 \|\epsilon - \hat{\epsilon}\|_2^2 \tag{5}$$

The denoising autoencoder reconstruction loss is optimized jointly with the one of the decoder:

$$\mathcal{L}_{\text{GEN}} = \mathcal{L}_{\text{AUX}}^{\text{rec}} + \mathcal{L}_{\text{REC}} \tag{6}$$

The pseudocode of the training procedure is shown in Algorithm 4 in Appendix C.1.

## 5 Counterfactual Generation

In the previous section we showed how to train a deep generative model with a Gaussian classifier that labels instances according to their label-relevant latent representation. Now we present our proposal to generate counterfactuals explaining the predictions to human users. With regard to the counterfactual search process, this only applies to label-relevant dimensions and we optimize latent distances under a validity constraint The underlying assumption is that optimization in the latent space will naturally translate to the input space. This alignment occurs when distances in the input space are accurately mirrored in the latent space, with reconstruction quality and the model's classification performance serving as reliable indicators of this condition. We start defining a set called counterfactual candidates whose elements optimize the trade-off betwee *likeliness* and *proximity* in the latent space. We then compute the expected value of these candidates according to the counterfactual class distribution and present it as the counterfactual explanation. This sidesteps the need for the user to specify (non-trivial) likelihood or distance thresholds for selecting the required counterfactual. To further enhance interpretability of the counterfactual explanation, we complement it with the most relevant concept changes. After training, concepts are extracted by human annotators in a post-hoc manner via latent traversals on the learned latent dimension. At explanation time, we return the concepts that were altered the most in generating the counterfactual (see Figure 1(a) for an illustration). These steps are further detailed in the following.

### 5.1 Counterfactual Candidates

We start by describing the formal properties of a candidate counterfactual while a graphical representation of the set of counterfactual candidates for an instance can be found in Figure 2(left).

**Definition 1** (properties of counterfactual candidates). *Let $x$ be an instance with encoding $z_0$ predicted as class $y^*$ with distribution centroid $\mu_{y^*}$. An instance $z_{cf}$ belongs to the set of counterfactual candidates $\mathcal{C}$ for the label $y_{cf}$ with centroid $\mu_{y_{cf}}$, if $\nexists z \neq z_{cf} \in \mathbb{R}^d$ that satisfies $\mathcal{P}_1 \wedge \mathcal{P}_2$, where:*

$$\mathcal{P}_1 : \underset{y}{\arg\min} \|z - \mu_y\|_2^2 = y_{cf}$$
$$\mathcal{P}_2 : \|z - z_0\|_2^2 \leq \|z_{cf} - z_0\|_2^2 \wedge \|z - \mu_{y_{cf}}\|_2^2 \leq \|z_{cf} - \mu_{y_{cf}}\|_2^2$$

$\mathcal{P}_1$ ensures the validity of the candidate counterfactual, i.e., the fact that it is always predicted as the alternative class. $\mathcal{P}_2$ ensures the non-existence of a strictly better counterfactual in the latent space.

It is straightforward to see that all the points that lie on the segment $\mathbb{S}_1$ from $z_0$ to $\mu_{y_{cf}}$ and satisfy the first condition are counterfactual candidates. These should be complemented with the points on the segment of the decision boundary $DB$ between class $y^*$ and $y_{cf}$ that goes from the intersection between $DB$ and $\mathbb{S}_1$ ($I_{cf}$) to the orthogonal projection of $z_0$ on $DB$ ($\text{PROJ}_{DB}(z_0)$).

**Proposition 1** (Set of counterfactual candidates). *Given an instance $x'$ with latent encoding $z_0$ predicted as class $y^*$, the set of counterfactual candidates $\mathcal{C}$ for label $y_{cf}$ consists of:*

1. *the points on the segment $\mathbb{S}_1$ from $z_0$ to $\mu_{y_{cf}}$ predicted as $y_{cf}$*

$$\mathbb{S}_1^{\mathcal{C}} = \{(1-t)z_0 + t\mu_{y_{cf}} \mid t \in [0,1] \wedge \mathcal{P}_1\} \tag{7}$$

2. *the points on the segment connecting the intersection $I_{cf}$ between $\mathbb{S}_1$ and the decision boundary $DB$ with the closest point to $z_0$ predicted as $y_{cf}$*

$$\mathbb{S}_2 = \{(1-t)I_{cf} + tPROJ_{DB}(z_0) \mid t \in [0,1]\} \tag{8}$$

Please refer to the Appendix B.1 for the proof. We proceed showing how to extract the expected counterfactual from this set of candidates.

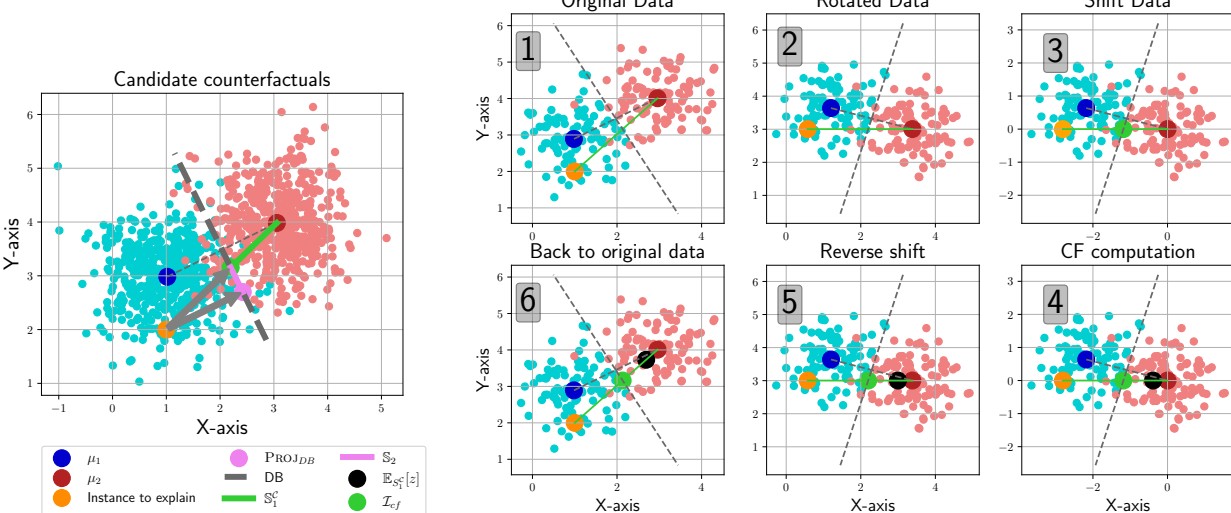

Figure 2: Visualisation of the set of candidates we take in consideration (left) and of the latent space manipulations necessary to compute the expected counterfactual (right).

## 5.2 Counterfactual as Expectation over Candidates

In the following section we define a technique to compute the expected value of the counterfactual candidates, which will be returned as a counterfactual explanation. We argue that such counterfactual intrinsically optimizes the trade-off between the likelihood of the explanation and the distance from the instance to explain in the latent space. More precisely, the expectation is a point of equilibrium as the weights of instances with more proximity and the weights of more likely instances balance out. Problematically, computing such expectation has no closed form solution, and a large number of samples from a multivariate normal distribution is necessary to estimate it. We thus derive specific conditions under which such estimate can be reduced to a fast and efficient sampling from a univariate distribution.

In our derivations we treat expected value computations separately for $\mathbb{S}_1^{\mathcal{C}}$ and $\mathbb{S}_2$, and return a density-based weighted sum of the two as the final counterfactual (more details in Appendix B.2.2):

$$z_{cf_1} = \mathbb{E}_{\mathbb{S}_1^{\mathcal{C}}}[z] \; ; \; z_{cf_2} = \mathbb{E}_{\mathbb{S}_2}[z] \; ; \; z_{cf} = w_1 z_{cf_1} + w_2 z_{cf_2}$$

$$\text{with } w_1 = \frac{\mathcal{N}(z_{cf_1}; \mu_{y_{cf}}, I)}{\mathcal{N}(z_{cf_1}; \mu_{y_{cf}}, I) + \mathcal{N}(z_{cf_2}; \mu_{y_{cf}}, I)} \text{ and } w_2 = 1 - w_1 \tag{9}$$

Methods like Monte Carlo Integration require a considerable number of samples to produce accurate estimates, since the density of points vanishes as the dimensions of the distributions increase. In order to speed-up the expected values estimation of equation 9, we propose an alternative sampling technique that achieves accurate results while being computationally efficient. More precisely, we note that if a generic segment $\mathbb{S}$ is parallel to a one of the axis, computing the corresponding expectation significantly simplifies.

**Proposition 2** (Expectation along a segment parallel to an axis)**.** *Let* $a = (c, c, ..., c, a_d)$ *and* $b = (c, c, ..., c, b_d) \in \mathbb{R}^d$ *be two points aligned along the last axis. Let* $\mathbb{S} = \{(1 - t)a + tb \mid t \in [0, 1]\}$ *be the segment connecting them, and* $Z(t) = (1 - t)a_d + t(b_d)$ *the function of the last component of the segment. In addition, let* $f_Z(z) = f_{Z_1, Z_2, ..., Z_d}(z)$ *be the density function of the underlying distribution of the expectation. The expected value of the elements in* $\mathbb{S}$ *according to an isotropic Gaussian is a vector with unchanged components except for the last one, computed as:*

$$\mathbb{E}_{\mathbb{S}}[z] = \left( c, c, ..., c, \int_0^1 Z(t) f_{Z_d}(Z(t)) dt \Big/ \int_0^1 f_{Z_d}(Z(t)) dt \right) \tag{10}$$

Please refer to Appendix B.2.1 for the proof. This expectation still has no closed form solution, but it is much cheaper to estimate as it requires univariate samples only. In the following we show how to extend this convenient result to segments that are not parallel to one of the axis.

### 5.3 Enabling One-Dimensional Sampling with Spatial Rotations

Unfortunately, segments $\mathbb{S}_1^{\mathcal{C}}$ and $\mathbb{S}_2$ are never simultaneously parallel to the last axis. However, rotating an isotropic Gaussian preserves the point densities, as distances are not affected by rotations. We can thus define a rotation matrix $R$ to map a generic segment $\mathbb{S}$ into a segment which is parallel to the last axis. More precisely, given $a$ and $b$ reference points in the space connected by a segment $\mathbb{S}$, we define a sequence of invertible rotations with respect to $m = \frac{a+b}{2}$, given the segment direction vector $v = b - a$. Each rotation will vanish the angle between the current component and the base vector of the next one, so to achieve our goal in $d - 1$ steps as depicted in Algorithm 1 (please refer to Appendix C.2 for additional details). To invert the rotations and map back to the original space, we simply store the rotation matrices and gradually update $z$ as $z \leftarrow R_i^T(z - m) + m$, where $R_i^T$ is the transpose of the rotations matrices presented in inverse order of computation. We name this inverse procedure ROTATE$^{-1}$. This procedure allows us to rotate the original label-relevant latent space, compute expectations with sampling on the rotated space, and map the expected value back to the original space without loss of information. This motivates embedding the latent space in a Gaussian-mixture, as other distributions would not allow to compute expectations via fast one-dimensional sampling. We now present the methodology we employ to boost the interpretability of proposed explanations through interpretable concept changes.

---

**Algorithm 1** Rotation Algorithm

ROTATE$(\cdot; m, v)$

**Input:** $m, v$, vector to map to rotated space $z$

1: $z^r \leftarrow z$
2: **for** $i = 0$ **to** $d - 1$ **do**
3:     $\theta \leftarrow \text{atan2}(v_i, v_{i+1})$
4:     $R \leftarrow I$
5:     $R_{i,i} \leftarrow \cos\theta$
6:     $R_{i,i+1} \leftarrow -\sin\theta$
7:     $R_{i+1,i} \leftarrow \sin\theta$
8:     $R_{i+1,i+1} \leftarrow \cos\theta$
9:     $z^r \leftarrow (z^r - m) \cdot R + m$
10: **end for**
11: **return** $z^r$

---

### 5.4 Concept-based Explanations

After training, we extract class-relevant concepts by traversing the latent space with each class medoid. This approach relies on a human annotator to identify the meaningful changes applied to input images when only a single dimension is altered at a time. Notably, this procedure enables the assignment of interpretable concepts to the model's learned latent representations without requiring direct supervision on these generative factors during training. One challenge is that constructing disentangled embedding spaces is not always feasible, as the problem is inherently unidentifiable without additional assumptions or supervision, which our approach deliberately avoids. To support our methodology, we conduct a qualitative analysis and present a latent traversal plot to illustrate how variations along individual latent dimensions, while keeping the others fixed, affect the model's generated output. Furthermore, the plot provides evidence that these changes are linked to interpretable concept variations in the model's generations. We encourage the reader to refer to Appendix F for further details. As result, during the counterfactual search step, we identify the top-$k$ most relevant latent dimensions for counterfactual generation and return the associated concepts. We quantify relevance score of a latent dimension as a likelihood-based squared difference:

**Definition 2.** *Let $x$ be an instance with latent encoding $z_0$ predicted as class $y^*$ with distribution centroid $\mu_{y^*}$. Let $z_{cf}$ be counterfactual encoding for an alternative class $y_{cf}$. Let $\mathbf{p}_y(z) = [\mathcal{N}(z_1; \mu_{y,1}, 1), \mathcal{N}(z_2; \mu_{y,2}, 1), \ldots, \mathcal{N}(z_d; \mu_{y,d}, 1)]$ be a vector of univariate densities for the single latent dimensions of $z$ according to a label $y$. Let $\Phi(y, z) = z \odot \mathbf{p}_y(z)$ be the Hadamard product between latent dimensions and their label-specific densities. The relevance scores of the latent dimensions for the counterfactual explanation are computed as follows:*

$$s_{cf} = (\Phi(y^*, z_0) - \Phi(y_{cf}, z_{cf})) \odot (\Phi(y^*, z_0) - \Phi(y_{cf}, z_{cf})) \tag{11}$$

The relevance score consists in the weighted squared differences between original and counterfactual encodings along each dimension. More precisely, each latent of the original encoding is weighted by its likelihood according to the predicted label distribution and each latent of the counterfactual encoding is weighted by its likelihood according to the counterfactual class distribution. This ensures that out-of-distribution components of the instance to explain do not affect too heavily the automatic process of concept retrieval, which could reduce clarity of the textual components of explanations. We finally return the top-$k$ most relevant concept changes associated to the top-$k$ latent dimensions in terms of relevance scores.

### 5.5 The Counterfactual Generation Algorithm

In the following section we assemble the various components presented so far into the full counterfactual generation process, presented in Algorithm 2. Given an instance $x$ predicted as having label $y^*$ and a user-provided counterfactual label $y_{cf} \neq y^*$, the explanatory pipeline consists of: 1) encoding the instance to explain $x$ in $z_s$ and $z_u$; 2) rotating the $\mathbb{S}_1^{\mathcal{C}}$ and $\mathbb{S}_2$ segments to align them on the last axis and sampling their expectations; 3) computing the expected counterfactual $z_{cf}$ in latent space by averaging the expectations from the segments; 4) Extracting top-$k$ most relevant concept changes, 5) concatenating the label-relevant and label-irrelevant latent representations and decoding the resulting latent vector into the final counterfactual explanation $x_{cf}$.

This procedure ensures explanations naturally connect to the original instance by sharing label-irrelevant factors, maintaining proximity. Efficient expected value estimation via sampling guarantees in-distribution outputs, and linking visual explanations to concept changes enhances interpretability, allowing users to focus on the relevant components of the explanation.

## 6 Experiments

### 6.1 Quantitative Evaluation

We quantitatively assess the quality of counterfactuals generated for the BloodMNIST dataset by our proposed framework and we carry an ablation study comparing with versions of our approach with missing

**Algorithm 2** Explanation Algorithm

**Input:** $x, y^*, y_{cf}, k$, instance to explain, prediction, counterfactual class and number of concepts.

**Encode instances and extract label relevant and label irrelevant encodings**

1: $z_s \leftarrow \text{ENC}_s(x)$

2: $z_u \leftarrow \text{ENC}_u(x)$

**Rotate space to compute expectations along $\mathbb{S}_1^{\mathcal{C}}$ and $\mathbb{S}_2$ sets of candidate counterfactuals**

3: $m_1 \leftarrow (z_s + \mu_{y_{cf}})/2; \ v_1 \leftarrow (\mu_{y_{cf}} - z_s)$

4: $S_1 \leftarrow \{(1-t)\text{ROTATE}(\mu_{y_{cf}}; m_1, v_1) + t\text{ROTATE}(z_s; m_1, v_1)\} \mid t \in [0,1] \wedge \mathcal{P}_1\}$

5: $z_{cf_1} \leftarrow \text{ROTATE}^{-1}(\mathbb{E}_{S_1^{\mathcal{C}}}[z]; m_1)$

6: $m_2 \leftarrow (\mu_{y^*} + \mu_{y_{cf}})/2; \ v_2 \leftarrow (\mu_{y_{cf}} - \mu_{y^*})$

7: $S_2 \leftarrow \{(1-t)\text{ROTATE}(z_s; m_2, v_2) + t\text{ROTATE}(\text{proj}_P(z_s); m_2, v_2)\} \mid t \in [0,1]\}$

8: $z_{cf_2} \leftarrow \text{ROTATE}^{-1}(\mathbb{E}_{S_2}[z]; m_2)$

**Compute expected counterfactual as density based weighted sum**

9: $w_1 \leftarrow \mathcal{N}(cf_1; \mu_{y_{cf}}, I)/(\mathcal{N}(cf_1; \mu_{y_{cf}}, I) + \mathcal{N}(cf_2; \mu_{y_{cf}}, I))$

10: $z_{cf} \leftarrow w_1 z_{cf_1} + (1 - w_1)z_{cf_2}$

**Extract concepts according to relevance metric**

11: $s_{cf} \leftarrow (\Phi(y^*, z_s) - \Phi(y_{cf}, z_{cf})) \odot (\Phi(y^*, z_s) - \Phi(y_{cf}, z_{cf}))$

12: Concepts $\leftarrow \text{EXTRACT}(s_{cf}, k)$

**Concatenate latent dimensions and decode to generate the explanation**

13: $x_{cf} \leftarrow \text{DEC}(\mathcal{M}_{\text{AUX}}([z_u; z_{cf}]))$

14: **return** $x_{cf}$, Concepts

---

various components of the explanatory pipeline. In addition, we leverage the FID, COUT, and $S^3$ metrics to evaluate *Likeliness*, *validity* and *proximity* of generated explanations. Overall, our proposal delivers competitive performance. In conclusion, since our approach enables efficient counterfactual generation using a gradient-free optimization process, we present a generation-times comparison plot. The results demonstrate that our technique is more efficient, while other methods struggle to meet the real-time performance requirements necessary for user interaction. We invite the reader to Appendix D for a detailed illustration of the obtained results.

## 6.2 User Study

To the best of our knowledge, our proposal is the first interactive classification framework that leverages an interpretable counterfactual generating technique that operates without concept supervision while enabling real time collaboration with users Average generation time for a single counterfactual with our method is in-fact $1.214 \pm 0.045$ seconds and Gaussian classification ensures 100% validity on generated explanations, as it allows to exploit linear decision boundaries to identify candidates guaranteed to respect property 1 of Definition 1. In addition, we facilitate the interaction step by eliminating the need for hyper-parameter configuration, thereby reducing potential confusion for non-expert users. More precisely, while an alternative approach would be to generate explanations based on a predefined likelihood value, our method offers a more user-friendly solution by automatically identifying an optimal value for this hyperparameter. This eliminates the need for non-expert users to select it without a clear understanding of its impact on the generated explanations. For a detailed analysis of how this hyperparameter choice affects explanation quality, we invite readers to refer to Appendix D. For these reasons we consider a challenging human-machine classification task with real-time feedback from the machine counterpart the most natural test-bed for our proposal. In the following sections we present the experiment designed to assess the effectiveness of our explanations and present the corresponding empirical findings.

### 6.2.1 Study design

We design an experiment with the goal of answering the following research questions:

**RQ1:** Can explanations improve users performance in solving the task?

**RQ2:** Can users spot machine errors in presence of explanations?

**RQ3:** Do explanations cause more user misatkes?

We focused on a multiclass image classification task, namely identifying the cell type of a blood cell image, using the BloodMNIST dataset introduced by Yang et al. (2023). The task is very challenging for a non-expert human, because of the poor resolution of the images and the difficulty in clearly identifying distinctive patterns per-class. Figure 9 in Appendix F reports the medoid image for the eight different cell types in the dataset. We trained our model on a 70-10-20 train-validation-test split, coarsely optimizing the hyper-parameters on the validation set (Appendix E.1). The resulting classifier achieves 91% accuracy on the test-set. We extracted a subset of 20 images from the test set to be presented to the user in the study. To address **RQ2** and **RQ3** while maintaining a manageable number of questions for the user, we included in this subset five images where the model is wrong. The accuracy of the trained model users interact with is therefore 75%, while the average accuracy of non-expert users is 27%, as will be shown in the following.

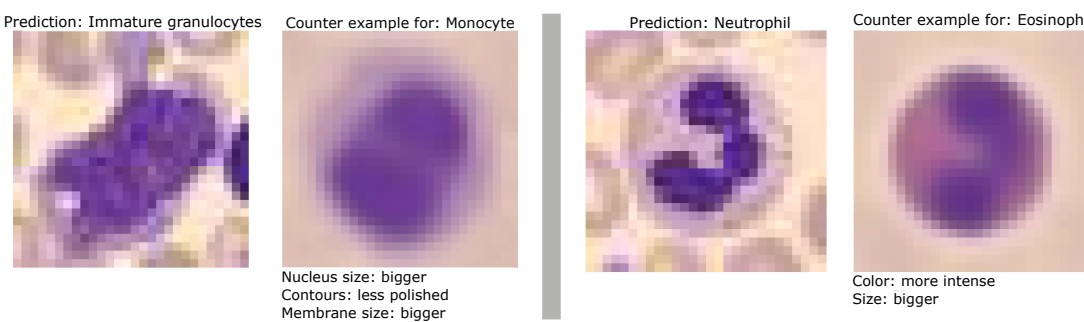

Figure 3: Examples of model prediction and counterfactual explanation for an alternative (user predicted) class. Concepts highlight the most relevant changes from the original image to the counterfactual.

We designed three experimental study variants to evaluate non-expert user performance in a cell type prediction task: no machine support (`None`), machine-predicted label (`Label`), and machine-predicted label with counterfactual explanation (`Label+Explanation`). Each variant involved 50 unique, English-speaking participants recruited via Prolific. Participants underwent brief preparatory training (Figure 15, Appendix H.4) before predicting the cell types of 20 test images. For each prediction, users were provided with the image and reference examples of all cell types (Figure 16, Appendix H.5). In `None`, participants received no machine feedback, serving as a baseline for human performance. In `Label`, users initially made their own predictions, as in `None`. If the machine disagreed, they were given the option to confirm their prediction, accept the machine's label, or select another. `Label+Explanation` extended `Label` by including a counterfactual explanation in case of disagreement: a counterfactual image resembling the original but predicted with the user-specified label, along with the top-3 concept changes required for this outcome (Figure 3). Additional details on the interface and study are in Appendix H.5.

### 6.2.2 Results

To answer our research questions we extract the following statistics: i) accuracy (ACC) before and after machine feedback, ii) agreement rate (AGR) with the machine before and after feedback, iii) accuracy against the machine (ACCAM), i.e., accuracy on instances where a user does not comply with the machine, iv) machine induced errors (MIE), namely errors made by users who initially provided correct answers, with respect to how many times the machine feedback altered their decisions. We invite readers to refer to Appendix H.1 for additional details on how ACCAM and MIE statistics are computed. All statistics are computed individually for each of the 50 study participants and Table 1 shows mean and standard deviation (sd) of the observed values. Results (Table 1) confirm the task's difficulty for non-experts, as participants in `None` struggled significantly. Notably, accuracy before feedback significantly improved in `Label` and `Label+Explanation` ($p$-values respectively of $5.275e^{-8}$ and $3.12e^{-14}$), suggesting that interacting with the machine provided implicit training (see Appendix H.3 ). Machine feedback also significantly boosted

| Types of feedback | ACCs (%) | | AGRs (%) | | ACCAM (%) | MIE (%) |
|---|---|---|---|---|---|---|
| | Before feedback | After feedback | Before feedback | After feedback | | |
| None | $26.73 \pm 8.46$ | - | - | - | - | - |
| Label | $50.60 \pm 12.19$ | $63.99 \pm 10.45$ | $\mathbf{43.90 \pm 9.96}$ | $70.80 \pm 13.97$ | $24.80 \pm 21.64$ | $\mathbf{16.24 \pm 13.20}$ |
| Label+Explanation | $\mathbf{51.63 \pm 11.06}$ | $\mathbf{69.08 \pm 8.39}$ | $41.96 \pm 10.55$ | $\mathbf{78.57 \pm 13.92}$ | $\mathbf{29.14 \pm 22.20}$ | $16.49 \pm 13.55$ |
| $p$-values | - | $\mathbf{0.004}$ | - | $\mathbf{0.003}$ | 0.163 | 0.463 |

Table 1: Comparison of users' performance across different settings. Reported values represent mean and sd of statistics computed individually for each of the 50 study participants.

overall accuracy, with the best results in `Label+Explanation`, where explanations helped up to 12% of users outperform the machine. The hypothesis that accuracy after feedback in `Label+Explanation` is higher than in `Label` was statistically confirmed via a t-test ($p = 0.004$). In addition, agreement rates were highest with explanations suggesting better trust and calibration of when to rely on feedback. The hypothesis that agreement after feedback in `Label+Explanation` is higher than in `Label` was statistically confirmed via a t-test ($p = 0.003$). Crucially, no evidence of over-reliance was observed, as users didn't alter correct predictions more often with explanations than without. The hypothesis that MIE in `Label+Explanation` is the same as in `Label` was not rejected in a t-test ($p = 0.463$). In conclusion, performance variability across participants highlights the overall task's complexity.

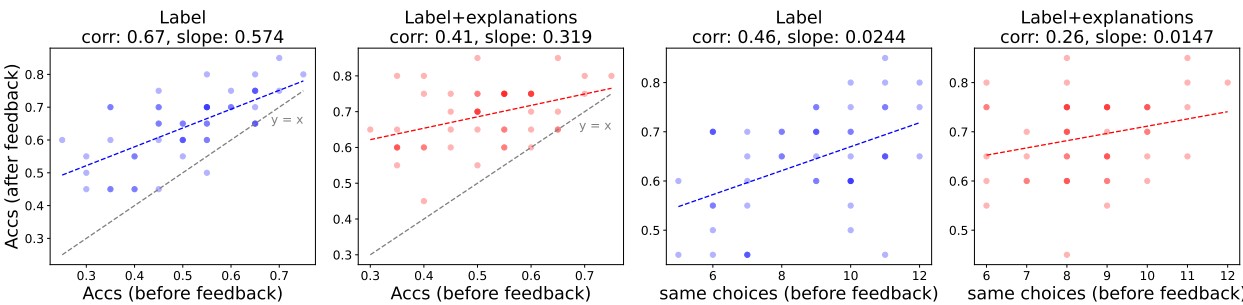

Figure 4: Comparison of correlation plots between the two settings of our experiment. Correlation significantly decrease in presence of explanations and slopes of regression lines become flatter.

We conclude investigating the relationship between a user final score ($ACC_{af}$) and their skill level, intended as accuracy before feedback ($ACC_{bf}$), as well as initial agreement ($AGR_{bf}$) which measures how many explanations a user is exposed to. Figure 4 shows correlation plots and Pearson's coefficients for the `Label` and `Label+Explanation` variants. Without explanations, $ACC_{bf}$ and $AGR_{bf}$ strongly predict final users scores as a consequence of the good performance of the machine. With explanations, this link weakens. Explanations seem to have the potential to flatten final scores, as the slope of regression lines suggest, therefore enabling users of varying skill levels to excel. See Appendix H.2 for a detailed discussion of feedback helpfulness across experimental settings.

In conclusion, our findings suggest affirmative answers to **RQ1** and a negative answer to **RQ3**. We highlight that, although a t-test fails to reject the hypothesis that ACCAM is the same in `Label+Explanation` and `Label` ($p = 0.163$), the slightly higher ACCAM value, combined with higher agreement levels, provides a weak but suggestive indication in favor of an affirmative answer to **RQ2**, albeit not conclusively. Additionally, despite considerable variance in user performance due to the complexity of the task, we can confidently assert that the explanations provided are beneficial across all user skill levels, demonstrating their overall utility.

## 7 Limitations and Future Work

The key requirement of our approach is that the latent space follows a multivariate Gaussian distribution. If this condition is not met, expectations cannot be efficiently computed using one-dimensional sampling. Applying our method in such cases may result in out-of-distribution generations and low-quality explanations, both in terms of *likeliness* and *interpretability*. Although this requirement can be met, using the Gaussian-Mixture Loss from Wan et al. (2018), it restricts the applicability of our approach to this specific class of models. Moreover, interpretable concepts traversal requires largely compressed latent spaces, as too complex structures can be challenging for users to comprehend, and this can hinder reconstruction quality for more complex input spaces. A potential solution is to condition latent diffusion models on RAE outputs to obtain refined counterfactuals, or directly on RAE's semantically meaningful latent representations. However, certain domains may not permit concept extraction, even with larger-scale models. Finally, we investigated the capabilities of our proposal within a single-stage interactive setting. Given that our approach is tailored for real-time collaboration, exploring potential improvements in *interpretability* through multi-stage interactions represents a significant future direction for our work. Exploring these directions while preserving the efficiency required for real-time interaction is an important avenue for future research.

## 8 Conclusion

We introduced the first framework for real-time interactive classification in the image domain, integrating interpretable counterfactual explanations. Our framework is built upon a novel explanatory approach that ensures the critical properties of *likeliness*, *validity*, and *proximity*, while also facilitating the efficient generation of counterfactuals. To validate our proposal, we conducted a user study assessing its effectiveness using a real-world dataset, where participants collaborated with the underlying model to perform a classification task. The results highlighted that explanations are beneficial for users of all skill levels, demonstrating the *interpretability* and practical utility of the provided machine feedback. Furthermore, the study underscored the essential role of *explanations* in enhancing user understanding and trust, showcasing their pivotal contribution to achieving clear and actionable insights.

### Broader Impact Statement

This study was conducted in compliance with the TMLR Code of Ethics. All participants provided informed consent before taking part in the study. The study involved the collection of anonymized data, ensuring that no personally identifiable information (PII) was recorded or stored at any point. Participants were informed about the purpose of the research, the voluntary nature of their participation, and their right to withdraw at any time without penalty. No sensitive personal information was collected, and all responses were kept confidential. The data were processed and analyzed solely for the purposes of this research and will not be used for any other purpose.

To facilitate the reproducibility of our results, we provide detailed information in the Appendix of this paper. This includes proofs of all propositions presented, a comprehensive description of the model architecture and of its training hyper-parameters and thorough explanations of all the algorithms used. Additionally, the Appendix contains information about the user study design and implementation. In conclusion, the source code of our implementation can be found at: `https://anonymous.4open.science/r/Interpretable-counterfactuals-real-time-C8D3/`. These efforts are intended to support researchers in replicating our methodology and verifying the robustness of our findings.

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

## A  Appendix

## B  Mathematical proofs

### B.1  Counterfactual Candidates

To better follow our proof, first let us introduce once again the properties of counterfactual candidates of Definition 1:

Let $x_0$ be an instance with encoding $z_0$ predicted as class $y^*$ with distribution centroid $\mu_{y^*}$. An instance $z_{cf}$ belongs to the set of counterfactual candidates $\mathbb{C}$ for the label $y_{cf}$ with centroid $\mu_{y_{cf}}$, if $\nexists\, z \neq z_{cf} \in \mathbb{R}^d$ that jointly satisfies $\mathcal{P}_1 \wedge \mathcal{P}_2$, where:

$$\mathcal{P}_1 : \underset{y}{\operatorname{argmin}} \|z - \mu_y\|_2^2 = y_{cf}$$

$$\mathcal{P}_2 : \|z - z_0\|_2^2 \leq \| z_{cf} - z_0\|_2^2 \wedge \|z - \mu_{y_{cf}}\|_2^2 \leq \| z_{cf} - \mu_{y_{cf}}\|_2^2$$

Counterfactual candidates should optimize a trade-off between *likeliness* and *proximity* under a *validity* constraint. More precisely, *likeliness* is measured as the euclidean distance between a point and the counterfactual class mean. The motivation is that, under diagonal covariance assumption $\Sigma = \sigma^2 I$, this distance is proportional to the negative log-likelihood according to the counterfactual class distribution:

$$\mathcal{N}(z, \mu, \sigma^2 I) = \frac{1}{(2\pi\sigma^2)^{\frac{d}{2}}} \exp\left(-\frac{1}{2\sigma^2}\|z - \mu\|_2^2\right)$$

$$-\log(\mathcal{N}(z, \mu, \sigma^2 I)) = \frac{1}{2\sigma^2}\|z - \mu\|_2^2 + c \propto \|z - \mu\|_2^2$$

According to the definition we provided, identifying candidates is trivial with the use of triangle inequality. Follows that all points satisfying $\mathcal{P}_1$ and laying on the segment $\mathbb{S}_1$ from $z_0$ to $\mu_{y_{cf}}$ are counterfactual candidates. This allows to omit the majority of points in space that satisfy the first property in favor of a point in $\mathbb{S}_1$. Problematically, some points in $\mathbb{S}_1$ are not predicted as the counterfactual class. This allows the existence of valid candidates according to $\mathcal{P}_1$ that cannot be discarded because they are equivalently distant from $z_0$ with respect to some points in $\mathbb{S}_1$ that do not satisfy $\mathcal{P}_1$. In the following we prove that when this happens an infinitesimal approximation of the best possible valid points according to $\mathcal{P}_2$ is obtained with the segment $\mathbb{S}_2$. This is the part of the decision boundary $DB$ between class $y^*$ and $y_{cf}$ that goes from the intersection between $DB$ and $\mathbb{S}_1$ ($I_{cf}$) to the orthogonal projection of $z_0$ on $DB$ ($\textsc{Proj}_{DB}(z_0)$). More precisely we define segments $\mathbb{S}1$ and $\mathbb{S}_2$ as below:

$$\mathbb{S}_1 = \{(1-t)z_0 + t\mu_{y_{cf}} \mid t \in [0,1]\}$$

$$\mathbb{S}_2 = \{(1-t)I_{cf} + t\textsc{Proj}_{DB}(z_0) \mid t \in [0,1]\}$$

And our proof is structured as follows:

1. We identify the set of points in $\mathbb{S}_1$ that are at least as distant to $z_0$ as $\textsc{Proj}_{DB}(z_0)$ but fail to satisfy $\mathcal{P}_1$, which we name $\mathbb{S}_1^{\mathcal{Q}}$.

2. For any point $z_S \in \mathbb{S}_1^{\mathcal{Q}}$ we construct the set of points $\mathbb{Z}_{DB}$ where $z_{DB} \in \mathbb{Z}_{DB}$ if $z_{DB} \in DB$ and $\|z_S - z_0\|_2^2 = \|z_{DB} - z_0\|_2^2$

3. We identify the best point $z_{DB}^* \in \mathbb{Z}_{DB}$ according to $\mathcal{P}_2$

4. We show that this point belongs to $\mathbb{S}_2$

5. We identify the region of space $\mathbb{O}$ containing the points that are better than $z_{DB}^*$ according to $\mathcal{P}_2$

6. We show that the points in $\mathbb{O}$ are all on the same side of the decision boundary

7. We prove this side is not associated to counterfactual class prediction.

The last point allows us to conclude that, for the given value of $\|z_S - z_0\|_2^2$, no valid point according to $\mathcal{P}_1$ is better than $z_{DB}^*$ according to $\mathcal{P}_2$. Therefore $z_{DB}^* \in \mathbb{C}$. In the following we further detail the different steps of the proof.

### B.1.1 Definition of $\mathbb{S}_1^{\mathcal{O}}$

To begin our proof let us consider the following setting. Let $\mu_{y^*}$ and $\mu_{y_{cf}}$ be the mean vectors of the original and ==counterfactual label distributions== in the latent space respectively. The segment $\mathbb{S}_\mu$ is the segment connecting them. The decision boundary $DB$ between the two according to diagonal covariance matrix assumption $\Sigma = \sigma^2 I$ is a hyper-plane perpendicular to $\mathbb{S}_\mu$. Finally the intercept $I_\mu$ between $\mathbb{S}_\mu$ and $DB$ is given by: $I_\mu = \frac{\mu_{y_{cf}} + \mu_{y^*}}{2}$. According to our setting we define the segment $\mathbb{S}_1^{\mathcal{O}}$ as follows:

$$\mathbb{S}_1^{\mathcal{O}} = \{z_S \in \mathbb{S}_1 : \|z_S - z_0\|_2^2 < \|I_{cf} - z_0\|_2^2 \wedge \|z_S - z_0\|_2^2 > \|\mathrm{PROJ}_{DB}(z_0) - z_s\|_2^2\} \tag{12}$$

Intuitively, any point $z$ that satisfies $\mathcal{P}_1$ must be at least at distance $\|\mathrm{PROJ}_{DB}(z_0) - z_s\|_2^2$ to $z_0$ as $\mathrm{PROJ}_{DB}(z_0)$ is the closest point in $DB$ to $z_0$. In addition, if $\|z_s - z_0\|_2^2 < \|I_{cf} - z_0\|_2^2$ the point $z_s \in \mathbb{S}_1$ does not satisfy $\mathcal{P}_1$.

### B.1.2 Definition of points on the Decision Boundary for a given $z_S \in \mathbb{S}_1^{\mathcal{O}}$

Let us denote by $\mathbb{H}(z_a, z_b)$ the hyperspherical set of points $z : \|z - z_a\|_2^2 = \|z_b - z_a\|_2^2$. Also, for any point $z_S \in \mathbb{S}_1^{\mathcal{O}}$, all the points $z : \|z - z_0\|_2^2 = \|z_s - z_0\|_2^2$ lay on a hyper-sphere. Let us denote $\mathbb{Z}_{DB}(\mathbb{K})$ the intersection between the collection of points in the set $\mathbb{K}$ and $DB$: $\mathbb{Z}_{DB}(\mathbb{K}) = \mathbb{K} \cap DB$. Let us now fix a value for $z_S$. We can denote the set of points $z_{DB}$ that belong to $DB$ and are equally distant to $z_0$ as $z_S$ as follows:

$$\mathbb{Z}_{DB}^{z_0} = \mathbb{Z}_{DB}(\mathbb{H}(z_0, z_s))$$

### B.1.3 Optimal $z_{DB}^*$ according to $\mathcal{P}_2$

Let us define the points in $\mathbb{H}(\mu_{y_{cf}}, z_S)$ that belong to $DB$:

$$\mathbb{Z}_{DB}^{y_{cf}} = \mathbb{Z}_{DB}(\mathbb{H}(\mu_{y_{cf}}, z_S))$$

According to $\mathcal{P}_2$, the best point $z_{DB}^* \in \mathbb{Z}_{DB}^{z_0}$, as all points in $\mathbb{Z}_{DB}^{z_0}$ are equally distant to $z_0$ by definition, is the one such that:

$$z_{DB}^* = \underset{z_{DB} \in \mathbb{Z}_{DB}^{z_0}}{\mathrm{argmin}} \|z_{DB} - \mu_{y_{cf}}\|_2^2$$

In addition we have that if $\mathbb{Z}_{DB}^{y_{cf}*} = \mathbb{Z}_{DB}(\mathbb{H}(\mu_{y_{cf}}, z_{DB}^*))$ , then :

$$|\mathbb{Z}_{DB}^{y_{cf}*} \cap \mathbb{Z}_{DB}^{z_0}| = 1 \tag{13}$$

More precisely, $\mathbb{Z}_{DB}^{z_0}$ and $\mathbb{Z}_{DB}^{y_{cf}}$ are hyper-spheres of $d - 1$ dimensions centered respectively in $\mathrm{PROJ}_{DB}(z_0)$ and $I_\mu$ because $DB \perp \mathbb{S}_\mu$. Since fixing $z_s$ is equivalent to fixing the radius $r_{z_o}$ of $\mathbb{Z}_{DB}^{z_0}$, we want to find the minimum $r_{y_{cf}}$ of $\mathbb{Z}_{DB}^{y_{cf}}$ such that $\mathbb{Z}_{DB}^{z_0} \cap \mathbb{Z}_{DB}^{y_{cf}} \neq \emptyset$. This leaves us with the trivial optimum radius $r_{y_{cf}}^*$ of $\mathbb{Z}_{DB}^{y_{cf}*}$ such that $\mathbb{Z}_{DB}^{z_0}$ is tangent to $\mathbb{Z}_{DB}^{y_{cf}*}$. The point of tangency is exactly $z_{DB}^*$.

### B.1.4 Proof that $z_{DB}^* \in \mathbb{S}_2$

We showed that the optimal $z_{DB*} \in \mathbb{Z}_{DB}^{z_0}$ is such that the two hyper-spheres of points on the decision boundary are tangent. We now show that the point $z_{DB}^*$ belongs to $\mathbb{S}_2$. More precisely, since the point where two hyper-spheres are tangent lays on the segment connecting the centroids, $z_{DB}^*$ will belong to the segment $\mathbb{S}_{tan}$ connecting $I_\mu$ and $\text{PROJ}_{DB}(z_0)$.

$$\mathbb{S}_{tan} = \{(1-t)\text{PROJ}_{DB}(z_0) + tI_\mu \mid t \in [0,1]\} \tag{14}$$

which is the segment on the decision boundary that collects all the values of $z$ such that two hyper-spheres $\mathbb{Z}_{DB}(\mathbb{H}(\mu_{y_{cf}}, z))$ and $\mathbb{Z}_{DB}(\mathbb{H}(z_0, z))$ are tangent. Moreover, the point $I_{cf}$ also belongs to $\mathbb{S}_{tan}$ as: 1) by definition it is on the decision boundary, 2) $\mathbb{H}(z_0, I_{cf})$ is tangent to $\mathbb{H}(\mu_{y_{cf}}, I_{cf})$. More precisely, the last condition ensures that $\mathbb{H}(z_0, I_{cf}) \cap \mathbb{H}(\mu_{y_{cf}}, I_{cf}) = I_{cf} \in DB$. This implies that $\mathbb{Z}_{DB}(\mathbb{H}(z_0, I_{cf})) \cap \mathbb{Z}_{DB}(\mathbb{H}(\mu_{y_{cf}}, I_{cf})) = I_{cf}$ and therefore $\mathbb{Z}_{DB}(\mathbb{H}(z_0, I_{cf}))$ is tangent to $\mathbb{Z}_{DB}(\mathbb{H}(\mu_{y_{cf}}, I_{cf}))$ in $I_{cf}$. Follows that if $I \in \mathbb{S}_{tan}$ then:

$$\mathbb{S}_{tan} = \{(1-t)\text{PROJ}_{DB}(z_0) + tI_{cf} \mid t \in [0,1]\} \cup \{(1-t)I_\mu + tI_{cf} \mid t \in [0,1]\}$$

or:

$$\mathbb{S}_{tan} = \mathbb{S}_2 \cup \{(1-t)I_\mu + tI_{cf} \mid t \in [0,1]\} \tag{15}$$

Finally, since $z_{DB}^* \in \mathbb{S}_{tan}$, then $z_{DB}^* \in \mathbb{S}_2$ as $\|z_{DB}^* - z_0\|_2^2 < \|I_{cf} - z_0\|_2^2$ and every element in the other component is at least distance $\|I_{cf} - z_0\|_2^2$ to $z_0$.

### B.1.5 Strictly better points than $z_{DB}^*$ according to $\mathcal{P}_2$

We showed that out of all the points in $\mathbb{Z}_{DB}^{z_0}$ the best possible choice according to $\mathcal{P}_2$ is $z_{DB}^* \in \mathbb{S}_2$. We now show how to find the region $\mathbb{O}$ of points that are better or equal than $z_{DB}^*$ according to $\mathcal{P}_2$ to prove that $\mathcal{P}_1$ is never true in this region. More precisely, the region of points that are simultaneously closer to $z_0$ and $\mu_{y_{cf}}$ than $z_{DB}^*$ is trivially identified as the intersection between the areas of the hyper-spheres $\mathbb{H}(z_0, z_{DB}^*)$ and $\mathbb{H}(\mu_{y_{cf}}, z_{DB}^*)$:

$$\begin{aligned}
\mathbb{A}_{z_0} &= \{z \in \mathbb{R}^d : \|z - z_0\|_2^2 \leq \|z_{DB}^* - z_0\|_2^2\} \\
\mathbb{A}_{y_{cf}} &= \{z \in \mathbb{R}^d : \|z - \mu_{f_{cf}}\|_2^2 \leq \|z_{DB}^* - \mu_{f_{cf}}\|_2^2\} \\
\mathbb{O} &= \mathbb{A}_{z_0} \cap \mathbb{A}_{y_{cf}}
\end{aligned} \tag{16}$$

In addition, $|\mathbb{O}| > 1$ since the two hyper-spheres are not tangent as $z_{DB}^* \notin \mathbb{S}_1$ which is the segment connecting $z_0$ and $\mu_{y_{cf}}$.

### B.1.6 Classification of $\mathbb{O}$

Given that any point that is an improvement to $z_{DB}^*$ is in $\mathbb{O}$, we show that the elements in this region are all on the same side of the decision boundary. If this holds, we can show that they are all predicted as a different label with respect to the counterfactual class and this would ensure that no better point than $z_{DB}^*$ that satisfies $\mathcal{P}_1$ exists. More precisely, to prove that all elements in $\mathbb{O}$ are on the same side of the decision boundary we need to prove that $DB$ does not intersect the region $\mathbb{O}$, as $DB$ is linear. To achieve this, given $\mathbb{O}_H^{z_0} = \mathbb{O} \cap \mathbb{H}(z_0, z_{DB}^*) \cap DB$ and $\mathbb{O}_H^{y_{cf}} = \mathbb{O} \cap \mathbb{H}(\mu_{y_{cf}}, z_{DB}^*) \cap DB$, we can equivalently show that: $|\mathbb{O}_H^{z_0} \cup \mathbb{O}_H^{y_{cf}}| = 1$ or that $DB$ touches the two hyper-spheres in the region $\mathbb{O}$ in a single shared point and therefore does not intersect it. In that regard, remind that $\mathbb{Z}_{DB}^{z_0}$ and $\mathbb{Z}_{DB}^{y_{cf}*}$ are the intersections with the decision boundary of $\mathbb{H}(z_0, z_S)$ and $\mathbb{H}(\mu_{y_{cf}}, z_{DB}^*)$. It is trivial to see that $\mathbb{O}_H^{z_0} = \mathbb{O} \cap \mathbb{Z}_{DB}^{z_0}$ and $\mathbb{O}_H^{y_{cf}} = \mathbb{O} \cap \mathbb{Z}_{DB}^{y_{cf}*}$. Given that $z_{DB}^* \in \mathbb{O}_H^{z_0}$ and $z_{DB}^* \in \mathbb{O}_H^{y_{cf}}$, if all the points in $\mathbb{O}$ are better or equal to $z_{DB}^*$ according to $\mathcal{P}_2$ then $\mathbb{O} \cap \mathbb{Z}_{DB}^{z_0} = \mathbb{O} \cap \mathbb{Z}_{DB}^{y_{cf}*} = z_{DB}^*$ as $z_{DB}^*$ optimizes $\mathcal{P}_2$ for $\mathbb{Z}_{DB}^{z_0}$. This allows to conclude that:

$$\mathbb{O}_H^{z_0} = \mathbb{O}_H^{y_{cf}} = \{z_{DB}^*\} \tag{17}$$

$$|\mathbb{O}_H^{z_0} \cup \mathbb{O}_H^{y_{cf}}| = 1 \tag{18}$$

or that all elements in $\mathbb{O}$ are assigned the same class label by the model.

### B.1.7 Proof $z_{DB}^* \in \mathbb{C}$

Since the points in $\mathbb{O}$ all share the same model prediction, we conclude our proof by taking a point inside $\mathbb{O}$ for which we know the model decision. This allows us to extend that same decision to all points in $\mathbb{O}$. More specifically, as $\mathbb{O}$ contains all the points that are better or equal to $z_{DB}^*$ according to $\mathcal{P}_2$, the original point $z_S \in \mathbb{S}_1$ that violets $\mathcal{P}_1$ will belong to $\mathbb{O}$. This is because $z_S$ is equivalently distant to $z_0$ while according to triangle inequality being closer to $\mu_{cf}$. This proves that all points in $\mathbb{O}$ are not predicted as the counterfactual class and violate $\mathcal{P}_1$. We conclude that $\nexists z \neq z_{DB}^* \in \mathbb{R}^d : \mathcal{P}_1 \wedge \mathcal{P}_2$ or $z_{DB}^*$ is a counterfactual candidate:

$$z_{DB}^* \in \mathbb{C} \tag{19}$$

### B.1.8 On the Validity of points in $\mathbb{S}_2$

We are aware that points on the decision boundary are technically a violation of $\mathcal{P}_1$. Even though this is true, we still consider them as an infinitesimal approximation of the points that would change the model prediction. Simplifying further our setting, let $\mu_{y_{cf}} = (c, c, ..., c, \mu_{y^*,d})$ and $\mu_{y_{cf}} = (c, c, ..., c, \mu_{y_{cf},d})$ the mean vectors of the original label distribution and the counterfactual class distribution. The segment $\mathbb{S}_\mu$ connecting them is parallel to the last axis: $\mathbb{S}_\mu \parallel e^{(d)}$ where $e^{(d)}$ is the basis vector of the last dimension. The decision boundary $DB$ between the two according to identity covariance matrix assumption is a hyper-plane perpendicular to $\mathbb{S}_\mu$: $DB \perp \mathbb{S}_\mu$. Finally the intercept $I_\mu$ between $\mathbb{S}_\mu$ and $DB$ is given by: $I_\mu = (c, c, ..., c, \frac{\mu_{y_{cf},d} + \mu_{y^*,d}}{2})$. According to this setting we have:

$$f_{\mathcal{M}}(z_{DB}^* + \epsilon e^d) = y_{cf} \text{ for } \epsilon \approx 0, \epsilon \in \mathbb{R}^+$$

As a global result, any infinitesimal change perpendicular to the decision boundary would result in the model predicting the counterfactual label.

## B.2 Expected Counterfactual

In the following we present mathematical derivations regarding the computation of the expected counterfactual.

### B.2.1 Expectation along a segment parallel to an axis

We show that the expected value of elements in a segment $S$, which lies parallel to the last axis, can be computed using single-dimensional sampling (as depicted by equation 10), assuming the elements belong to a space $\mathbb{R}^d$ where they follow an isotropic Gaussian distribution:

$$\mathbb{E}_S[z] = \left( c, c, ..., c, \int_0^1 Z(t) f_{Z_d}(Z(t)) dt \bigg/ \int_0^1 f_{Z_d}(Z(t)) dt \right)$$

**proof**: Take two points aligned along the last axis $a = (c, c, ..., c, a_d)$ and $b = (c, c, ..., c, b_d) \in \mathbb{R}^d$, with $c, a_d, b_d \in \mathbb{R}$ and $a_d < b_d$ and the segment $S$ connecting them $S = \{(1-t)a + (t)(b) \mid t \in [0,1]\}$. Any point $z \in S$ can be expressed as a function of $t$: $Z(t) = (1-t)a + (t)(b)$. More precisely any coordinate of any point $z \in S$ can be expressed as a function of the corresponding components of $a$ and $b$ and $t$: $Z_i(t) = (1-t)a_i + t(b_i)$. If the underlying distribution of the points in $S$ is an isotropic Gaussian we can factorize the density as follows:

$$f_{Z_1,...,Z_d}(z_1, ..., z_d) = \prod_i^d f_{Z_i}(z_i)$$

And the expected value becomes:

$$\mathbb{E}_S[z] = \frac{\int_0^1 Z(t) f_Z(Z(t)) dt}{\int_0^1 f_Z(Z(t)) dt} = \frac{\int_0^1 Z(t) \prod_{i=1}^d f_{Z_i}(Z_i(t)) dt}{\int_0^1 \prod_{i=1}^d f_{Z_i}(Z_i(t)) dt}$$

But:

$$\prod_{i=1}^{d} f_{Z_i}(Z_i(t)) = f_{Z_d}(Z_d(t)) \prod_{i=1}^{d-1} f_{Z_i}(c)$$

and:

$$\frac{\int_0^1 Z(t) \prod_{i=1}^{d} f_{Z_i}(Z_i(t))dt}{\int_0^1 \prod_{i=1}^{d} f_{Z_i}(Z_i(t))dt} = \frac{\prod_{i=1}^{d-1} f_{Z_i}(c) \int_0^1 Z(t) f_{Z_d}(Z_d(t))dt}{\prod_{i=1}^{d-1} f_{Z_i}(c) \int_0^1 f_{Z_d}(Z_d(t))dt} = \frac{\int_0^1 Z(t) f_{Z_d}(Z_d(t))dt}{\int_0^1 f_{Z_d}(Z_d(t))dt}$$

To conclude our proof we have that for a given $t$ value $Z(t)$ is a vector of the form $(c, c, ..., c, Z_d(t))$ and we can write:

$$\mathbb{E}_S[z] = \left( \frac{\int_0^1 c f_{Z_d}(Z_d(t))}{\int_0^1 f_{Z_d}(Z_d(t))dt}, ..., \frac{\int_0^1 c f_{Z_d}(Z_d(t))}{\int_0^1 f_{Z_d}(Z_d(t))dt}, \frac{\int_0^1 Z_d(t) f_{Z_d}(Z_d(t))dt}{\int_0^1 f_{Z_d}(Z_d(t))dt} \right)$$

$$= \left( \frac{c \int_0^1 f_{Z_d}(Z_d(t))}{\int_0^1 f_{Z_d}(Z_d(t))dt}, ..., \frac{c \int_0^1 f_{Z_d}(Z_d(t))}{\int_0^1 f_{Z_d}(Z_d(t))dt}, \frac{\int_0^1 Z_d(t) f_{Z_d}(Z_d(t))dt}{\int_0^1 f_{Z_d}(Z_d(t))dt} \right)$$

$$= \left( c, ..., c, \frac{\int_0^1 Z_d(t) f_{Z_d}(Z_d(t))dt}{\int_0^1 f_{Z_d}(Z_d(t))dt} \right)$$

Proving that to estimate the last component, which is the only one whose value is modified, we can resort to one-dimensional sampling.

In conclusion, the clear advantage is that eliminating other dimensions significantly increases the probability of sampling within the desired interval removing the complexity of combinatorial effects. More precisely, dimensionality has no influence on the effectiveness of our approach, whereas it poses a problem for other sampling-based methods, as it causes probability densities to vanish due to factorization.

### B.2.2 Expected Candidate Computation

Given two generic segments $\mathbb{S}_1 = \{(1-t)a_1 + (t)(b_1) \mid t \in [0,1]\}$ and $\mathbb{S}_2 = \{(1-t)a_2 + (t)(b_2) \mid t \in [0,1]\}$ and $a_1, b_1, a_2, b_2 \in \mathbb{R}^d$, The expected value of elements in the segments equals:

$$\mathbb{E}_{\mathbb{S}_1, \mathbb{S}_2}[z] = w_1 \mathbb{E}_{\mathbb{S}_1}[z] + w_2 \mathbb{E}_{\mathbb{S}_2}[z]$$

$$\text{with } w_1 = \frac{\int_0^1 f_Z(Z_1(t))dt}{\int_0^1 f_Z(Z_1(t))dt + \int_0^1 f_Z(Z_2(t))dt} \text{ and } w_2 = 1 - w_1$$

$$\text{where } Z_1(t) = (1-t)a_1 + tb_1 \text{ and } Z_2(t) = (1-t)a_2 + tb_2$$

This formulation requires an additional Monte-Carlo estimator of the probabilities of the segments and for efficiency in our derivations we approximate the quantity with:

$$z_1 = \mathbb{E}_{\mathbb{S}_1}[z] \; ; \; z_2 = \mathbb{E}_{\mathbb{S}_2}[z] \; ; \; z = w_1 z_1 + w_2 z_2$$

$$\text{with } w_1 = \frac{\mathcal{N}(z_1; \mu_{y_1}, I)}{\mathcal{N}(z_1; \mu_{y_1}, I) + \mathcal{N}(z_2; \mu_{y_1}, I)} \text{ and } w_2 = 1 - w_1$$

It is worth noticing that in our setting we would have $\mathcal{N}(Z_1(t); \mu, I) > \mathcal{N}(Z_2(t); \mu, I) \, \forall t \in [0,1]$ therefore:

$$\int_0^1 f_Z(Z_1(t))dt \gg \int_0^1 f_Z(Z_2(t))dt$$

which inevitably transfers to the mean densities:

$$\mathcal{N}(z_1; \mu_{y_1}, I) \gg \mathcal{N}(z_2; \mu_{y_1}, I)$$

Thus, we can conclude that the approximation for the expected value is suitable:

$$\frac{\displaystyle\int_0^1 f_Z(Z_1(t))dt}{\displaystyle\int_0^1 f_Z(Z_1(t))dt + \int_0^1 f_Z(Z_2(t))dt} \approx \frac{\mathcal{N}(z_1; \mu_{y_1}, I)}{\mathcal{N}(z_1; \mu_{y_1}, I) + \mathcal{N}(z_2; \mu_{y_1}, I)} \tag{20}$$

### B.2.3 Expected Counterfactual Violations of $\mathcal{P}_2$

The expected counterfactual can violate the second property of counterfactual candidates defined as:

$$\mathcal{P}_2 : \|z - z_0\|_2^2 \le \|z_{cf} - z_0\|_2^2 \wedge \|z - \mu_{y_{cf}}\|_2^2 \le \|z_{cf} - \mu_{y_{cf}}\|_2^2$$

This is because the expected counterfactual consists in an interpolation of points in $\mathbb{S}_1^{\mathcal{C}}$ and $\mathbb{S}_2$ which inevitably returns a point that belongs to neither segment. Given a generic segment $\mathbb{S} = \{(1-t)a + (t)(b) \mid t \in [0,1]\}$ with $a, b \in \mathbb{R}^d$ and two additional points $c = t_0 a + (1-t_0)b$ that belongs to $\mathbb{S}$ and $d \in \mathbb{R}^d$ we define the interpolation between $c$ and $d$ as $c_1 = w_1 c + (1-w_1)d$. The distance between the interpolation $c_1$ and any point in the segment $\mathbb{S}$ is given by:

$$\| (1-t)a + (t)(b) - (1-t_0)a - (t_0)(b) - (1-w_1)d \|_2^2$$

which allows us to bound the distance between the interpolation $c_1$ and the segment $\mathbb{S}$ with at least:

$$\| (1-t_0)a + (t_0)(b) - (1-t_0)a - (t_0)(b) - (1-w_1)d \|_2^2$$
$$\| (1-w_1)d \|_2^2 = (1-w_1)^2 \| d \|_2^2 \tag{21}$$

Recall from 20 that the weight associated to the expected value of $\mathbb{S}_1^{\mathcal{C}}$ appraoches one implying that $1 - w_1$ approaches zero. This allows us to conclude that, while the expected counterfactual slightly violates the $\mathcal{P}_2$ property of counterfactual candidates, this violation is negligible due to the inherent relationship between $\mathbb{S}_1^{\mathcal{C}}$ and $\mathbb{S}_2$.

## C  Algorithms

### C.1  Training Algorithms

We minimize this loss of 4 following the procedure depicted in Algorithm 3. We encode inputs to extract label-relevant and label-irrelevant dimensions and compute the corresponding classification and regularization components of the loss. Follows that latents are concatenated and decoded to compute reconstruction loss before the update-step of model parameters. Procedure iterates until convergence.

**Algorithm 3** Deterministic Training

   **Procedure:** $\text{DetTrain}(\lambda_s, \lambda_u, n)$
  **while** not convergence **do**
    **for** $i = 0$ **to** $n$ **do**
      $\{x, y\} \sim \mathcal{D}$
      $z_s \leftarrow ENC_s(x)$
      $z_u \leftarrow ENC_u(x)$
      $\tilde{x} \leftarrow DEC([z_s; z_u])$
      $\mathcal{L} \leftarrow \mathcal{L}_{\text{REC}} + \lambda_s \mathcal{L}_{GM} + \lambda_u \mathcal{L}_{GM}^u$
      $\psi, \phi, \pi \overset{+}{\leftarrow} -\nabla_{\psi,\phi,\pi}\mathcal{L}$
    **end for**
  **end while**

**Algorithm 4** Generative Training

   **Procedure:** $\text{GenTrain}(\sigma, n)$
  **while** not convergence **do**
    **for** $i = 0$ **to** $n$ **do**
      $\{x, y\} \sim \mathcal{D}; \quad \epsilon \sim \mathcal{N}(0, I)$
      $z_s \leftarrow ENC_s(x) + \sigma \cdot \epsilon$
      $z_u \leftarrow ENC_u(x) + \sigma \cdot \epsilon$
      $z_{\text{aux}} \leftarrow ENC_{\text{AUX}}([z_s; z_u])$
      $\tilde{z} \leftarrow DEC_{\text{AUX}}(z_{\text{aux}})$
      $\tilde{x} \leftarrow DEC(\tilde{z})$
      $\mathcal{L} \leftarrow \mathcal{L}_{\text{AUX}}^{\text{rec}} + \mathcal{L}_{\text{REC}}$
      $\theta, \omega, \pi \overset{+}{\leftarrow} -\nabla_{\theta,\omega,\pi}\mathcal{L}$
    **end for**
  **end while**

The procedure of our second stage of training is depicted in Algorithm 4. We encode latent representations to extract label-relevant and label-irrelevant codes. Through reparametrization trick we inject noise to both representations. We now introduce our auxiliary model which takes as input the concatenation of these noisy latents and is trained to denoise them. We compute the auxiliary loss component as in equation 5 and reconstruct original inputs from the denoised representations. Finally the loss of 6 is computed and parameters updated. This procedure iterates until convergence.

## C.2 Rotation Algorithm

We describe the algorithm we use to rotate the space so that the segment $S$ connecting $z$ and $z'$ is parallel to the last-axis. More precisely, given inputs $z$ of dimensionality $d$, $v = z' - z$ direction vector and the reference point $m = (z + z')/2$ (left unchanged by rotations), our algorithm returns the point $z^r$ that corresponds to $z$ in the rotated space.

**Algorithm 5** Rotation Algorithm

$\text{Rotate}(\cdot; m, v)$
**Input:** $m, v$, vector to map to rotated space $z$
 1: $z^r \leftarrow z$
 2: **for** $i = 0$ **to** $d - 1$ **do**
 3:   $\theta \leftarrow \text{atan2}(v_i, v_{i+1})$
 4:   $R \leftarrow I$
 5:   $R_{i,i} \leftarrow \cos\theta$
 6:   $R_{i,i+1} \leftarrow -\sin\theta$
 7:   $R_{i+1,i} \leftarrow \sin\theta$
 8:   $R_{i+1,i+1} \leftarrow \cos\theta$
 9:   $z^r \leftarrow (z^r - m) \cdot R + m$
10: **end for**
11: **return** $z^r$

When a direction vector's components are all simultaneously zero except for the last one the vector becomes parallel to the last axis. Based on this observation, we define an iterative procedure that progressively zeros out each dimension and aligns the corresponding axis. Once the second-to-last dimension is processed, the vector will be fully parallel to the last axis and the procedure completed. More precisely, given a direction vector $v$, for each dimension $i$ we compute the angle $\theta$ between $v_i$ and $e^{(i+1)}$ using $\theta = \text{atan2}(v_i, v_{i+1})$, where $e^{(i+1)}$ is the basis vector of the $(i+1)$-th dimension. This angle defines the rotation needed to zero out the

current dimension. Once $\theta$ is computed, we construct a rotation matrix $R$ that affects only the $i$-th and $(i+1)$-th dimensions, leaving the rest unchanged. To achieve this we combine the identity matrix with the standard $2d$ rotation matrix for the indices of interest. The vector $z$ is then transformed by multiplying it with the rotation matrix $R$, effectively zeroing out the $i$-th dimension. This process is repeated iteratively for $d-1$ steps, progressively aligning the vector with the final axis.

# D Quantitative Evaluation

## D.1 Counterfactual Quality

In the following, we quantitatively assess the quality of counterfactuals generated for the BloodMNIST dataset by our proposed framework and competitors. As a baseline, we compare it to the method introduced by Luss et al. (2021), which, to the best of our knowledge, is the only other interpretable counterfactual generation framework that operates without concept supervision. In addition, to conduct an ablation study, we compare our approach with simpler approaches. The first approach generates counterfactuals by interpolating between the instance to be explained and the mean representation of the counterfactual class, ensuring the model's confidence reaches specific thresholds (0.6, 0.8, 0.9). The second approach discards label-irrelevant encoding, relying solely on a label-relevant encoder (SLR), which inevitably compromises the proximity of the generated explanations. We leverage the FID, COUT, and $S^3$ metrics to evaluate various desiderata of counterfactual explanations. The FID score (Heusel et al., 2017), typically used to evaluate the quality of generative models, quantifies the realism of the generated counterfactuals. The COUT score (Khorram & Fuxin, 2022) focuses on the confidence of the model in the original and counterfactual classes, providing insight into the effectiveness of the counterfactual explanation. Finally, the $S^3$ (Jeanneret et al., 2023) metric, which leverages the SimSiam self-supervised learning framework (Chen & He, 2021), compares the cosine similarities between the SimSiam encodings of the original and counterfactual instances.

| Method | FID | COUT | $S^3$ |
|---|---|---|---|
| OURS | **131.21** | 0.90 | 0.81 |
| CEM-MAF | 173.61 | 0.85 | **0.87** |
| Interpolation (0.6) | 264.79 | 0.22 | 0.63 |
| Interpolation (0.8) | 162.81 | 0.68 | 0.84 |
| Interpolation (0.9) | 135.44 | 0.83 | 0.81 |
| SLR | 133.15 | **0.94** | 0.69 |

Table 2: Comparison of counterfactual generation methods using various metrics to assess the likeliness, proximity, and impact of explanations on model confidence.

In Table 2 we present the methods along with their corresponding scores for each metric. While the FID score is relatively high across all methods, our approach achieves the best FID score. These high values are primarily due to the constrained latent spaces used by the methods, which produce counterfactuals that are clearly distinguishable from the original images. However, the results from our user study provide strong evidence that the generated counterfactuals are both actionable and informative. Our method also achieves the second-highest COUT score, surpassed only by the approach that models exclusively label-relevant dimensions. This is due to the ability of that approach to modify more latent dimensions during optimization, resulting in instances that are closer to the mean of the counterfactual label distribution. Overall results indicate that our approach generates impactful perturbations of the original instances so to achieve counterfactual explanations with high model confidence. The best $S^3$ score is achieved by CEM-MAF, which excels in this category due to its design focused on optimizing proximity. Overall, our approach delivers competitive performance, outperforming competitors in FID metric and obtaining valuable results on COUT score while performing slightly worse on the $S^3$ metric. Simpler approaches, as expected, show lower FID and COUT scores, although interpolation with a confidence threshold of 0.8 surpasses our method $S^3$ metric. The variability in the results of the interpolation approaches raises the question of what the model's confidence value should be, as it is difficult to generalize because this value depends on the model's learned decision

boundary. As a result, hyper-parameter tuning becomes a critical requirement for interpretability. Our approach, however, demonstrates better overall performance and eliminates the need for hyper-parameter tuning, making it a more favorable choice. This is particularly crucial in real-time user interaction settings, where automating the counterfactual generation process is essential.

## D.2 Generation Times

Our approach enables efficient counterfactual generation using a gradient-free optimization process, which offers a significant computational advantage over existing techniques. Specifically, the computational cost of our method depends solely on the dimensionality of the input latent vector, making the generation time independent of the complexity of the underlying model architecture. This contrasts with gradient-based optimization methods, where the depth of the model can dramatically slow down the convergence of the counterfactual generation process. In Table 3, we present a comparison of generation times between our method and the competing approach of (Luss et al., 2021). The results demonstrate that our technique is more efficient, while other methods struggle to meet the real-time performance requirements necessary for user interaction.

| Method | OURs | CEM-MAF (k values) | | |
|---|---|---|---|---|
| | | k=1 | k=3 | k=5 |
| **Generation time (s)** | **1.21 ± 0.05** | 15.87 ± 1.86 | 24.16 ± 11.05 | 31.08 ± 14.21 |

Table 3: Comparison of generation times for our method and CEM-MAF for different values of hyperparameter $k$ which controls the model confidence on the counterfactual prediction.

Table 3 shows the substantial efficiency gains offered by our approach, revealing that generation times are often insufficient, if not entirely inadequate, for providing real-time feedback, even when using basic and shallow neural network architectures. This issue is exacerbated in more complex domains as depicted in Figure 5 where generation times for different model architecture depths are compared. In contrast, our method preserves its efficiency independently from such complexities.

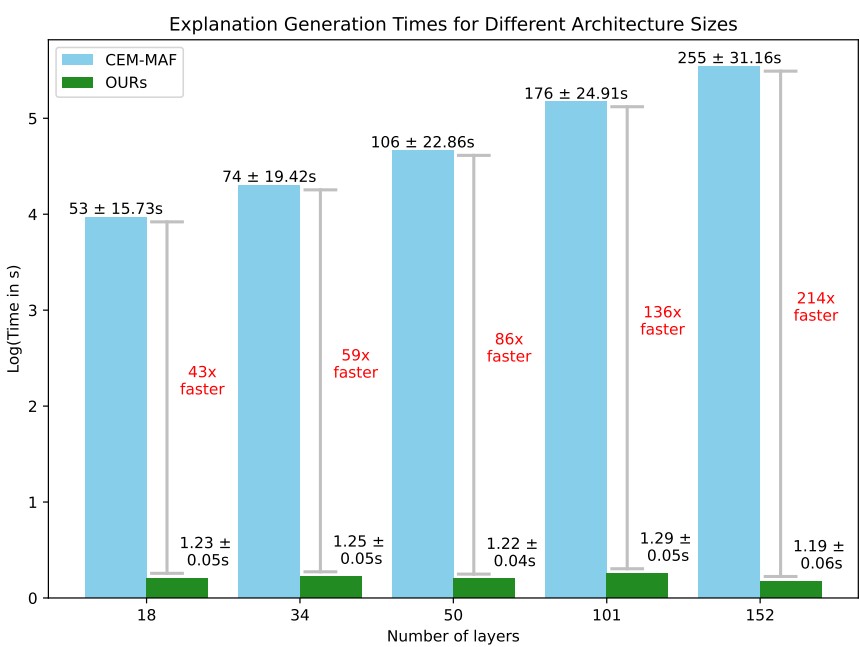

Figure 5: Comparison for generation times at varying of number of layers of a resnet architecture.

### D.3 Implementation Details

To implement the approach of Luss et al. (2021) we trained a Convolutional Neural Network classifier and Disentangled Inferred Priors Variational Autoencoder (Kumar et al., 2017) as their proposal suggests. The architectures of the two models were identical to the encoding and decoding blocks implemented for our Denoising Disentangled Regularized Autoencoder (Table 4) with the only exception that the classifier latent dimension was 8 (number of classes) and the DIP-VAE latent dimension was 10. In addition, we set all hyper-parameters as the proposed values in the popular repository `https://github.com/Trusted-AI/AIX360`. Specifically, the number of iterations was set to 250. If a valid counterfactual was not obtained within this limit, we permitted the algorithm to continue running until the first valid counterfactual was generated. The value of $k$ represents the difference in log-probabilities the model associates to the user asked class and the second most plausible class for the counterfactual explanation. The approach of Luss et al. (2021) returns explanations for which this difference is at least $k$ , with a common choice being $k = 5$. Intuitively, the optimization process slows down as the value of $k$ increases because achieving a higher model confidence in predicting a different class than the original necessitates progressively larger perturbations to the input.

## E Training

### E.1 Optimization and Architectures

We train our model on BloodMNIST dataset introduced by (Yang et al., 2023). It contains 17092 images of blood cells belonging to 8 different classes. We use a 70-10-20 train-validation-test split and optimize hyper parameters with the use of the validation set. For training, we use Adam optimizer with $\alpha = 0.001$, $\beta_1 = 0.9$ and $\beta_2 = 0.999$. With regard to the other hyper-parametrs, in the first stage of training we use $\lambda_s = 10, \lambda_u = 10$. The first was picked to avoid over-fitting by means of the validation set. With the second parameter we instead obtain a reasonable trade-off between learning meaningful high-level generative factors and adversarial classification performance. In the second stage of training we introduce noise according to $\sigma = 0.1$. More precisely, we empirically notice that a desirable trade-off between reconstruction quality and latent smoothing is obtained with this value. The factors that primarily affect this are learned latent-structure and size of latent space. Below we show architectures of the models implemented.

| Encoder | Decoder |
|---|---|
| input $x \in \mathbb{R}^{28 \times 3 \times 3}$ | input $x \in \mathbb{R}^{20}$ |
| 3x3 conv, 32 filters, batchnorm, relu | Dense 200 units, relu |
| 3x3 conv, 32 filters, batchnorm, relu | Dense 200 units, relu |
| 2x2 maxpool, stride 2 | Dense 8*8*64 units |
| 3x3 conv, 64 filters, batchnorm, relu | 3x3 trans conv, 64 filters, batchnorm, relu |
| 3x3 conv, 64 filters, batchnorm, relu | 3x3 trans conv, 64 filters, batchnorm, relu |
| Dense 200 units, relu | 2x2 upsample |
| Dense 200 units, relu | 3x3 trans conv, 32 filters, batchnorm, relu |
| Dense 15 for $z_s$, 5 for $z_u$ | 3x3 trans conv, 3 filters |

Table 4: Architecture for Encoder (ENC($\cdot$)) and Decoder (DEC($\cdot$))

| Auxiliary Encoder | Auxiliary Decoder |
|---|---|
| input $x \in \mathbb{R}^{20}$ | input $x \in \mathbb{R}^{12}$ |
| Dense 64 units, relu | Dense 16 units, relu |
| Dense 32 units, relu | Dense 32 units, relu |
| Dense 16 units, relu | Dense 64 units, relu |
| Dense 12 output units | Dense 20 output units |

Table 5: Architectures for auxiliary encoder (ENC$_{\text{AUX}}$($\cdot$)) and decoder (DEC$_{\text{AUX}}$()$\cdot$)

## E.2 Latent Space

Here we present the structure of the latent space learned by the model. as depicted in 6 the label-relevant dimensions are mapped to a ==label-relevant== space and class is indistinguishable according to label-irrelevant dimensions which follow an Isotropic Gaussian.

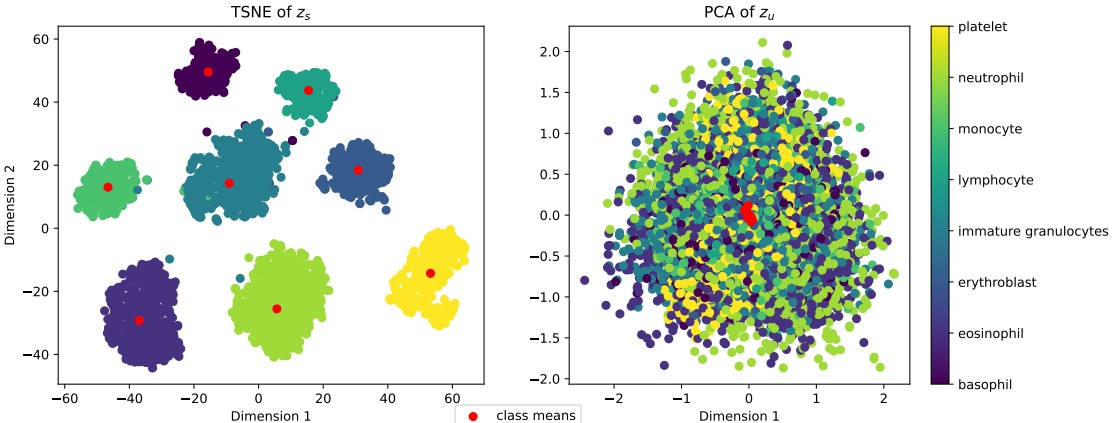

Figure 6: Learned latent structure. Gaussian mixture for label-relevant and isotropic gaussian for label-irrelevant dimensions.

## E.3 Sampling

After regularization with the noise injection mechanism, our model is suited for sampling. We extract distribution parameters for the label-relevant encodings and sample according to diagonal-covariance distributions. Label irrelevant encodings follow instead an isotropic gaussian. We show few examples of results with unconditional (Figure 7) and conditional sampling (Figure 8).

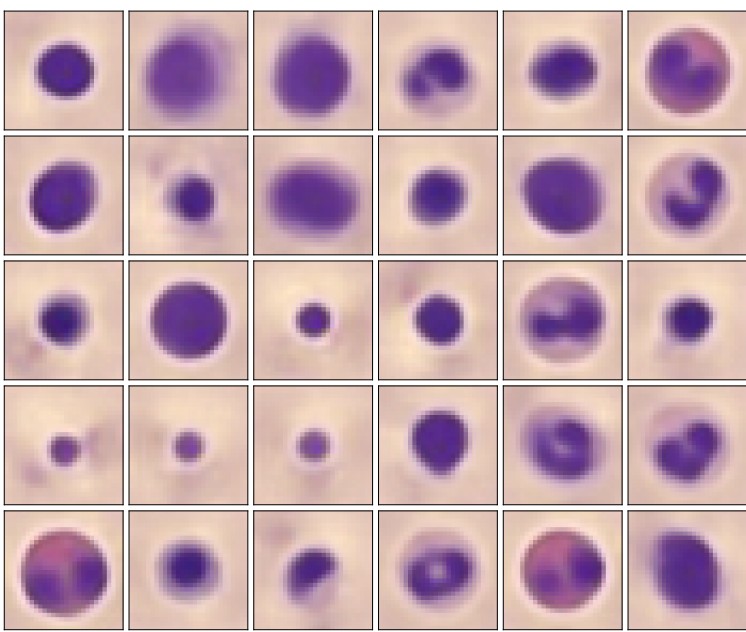

Figure 7: Unconditional sampling. To achieve this labels are treated as a random variable and sampled. Finally a new image is obtained from the conditional random label distribution.

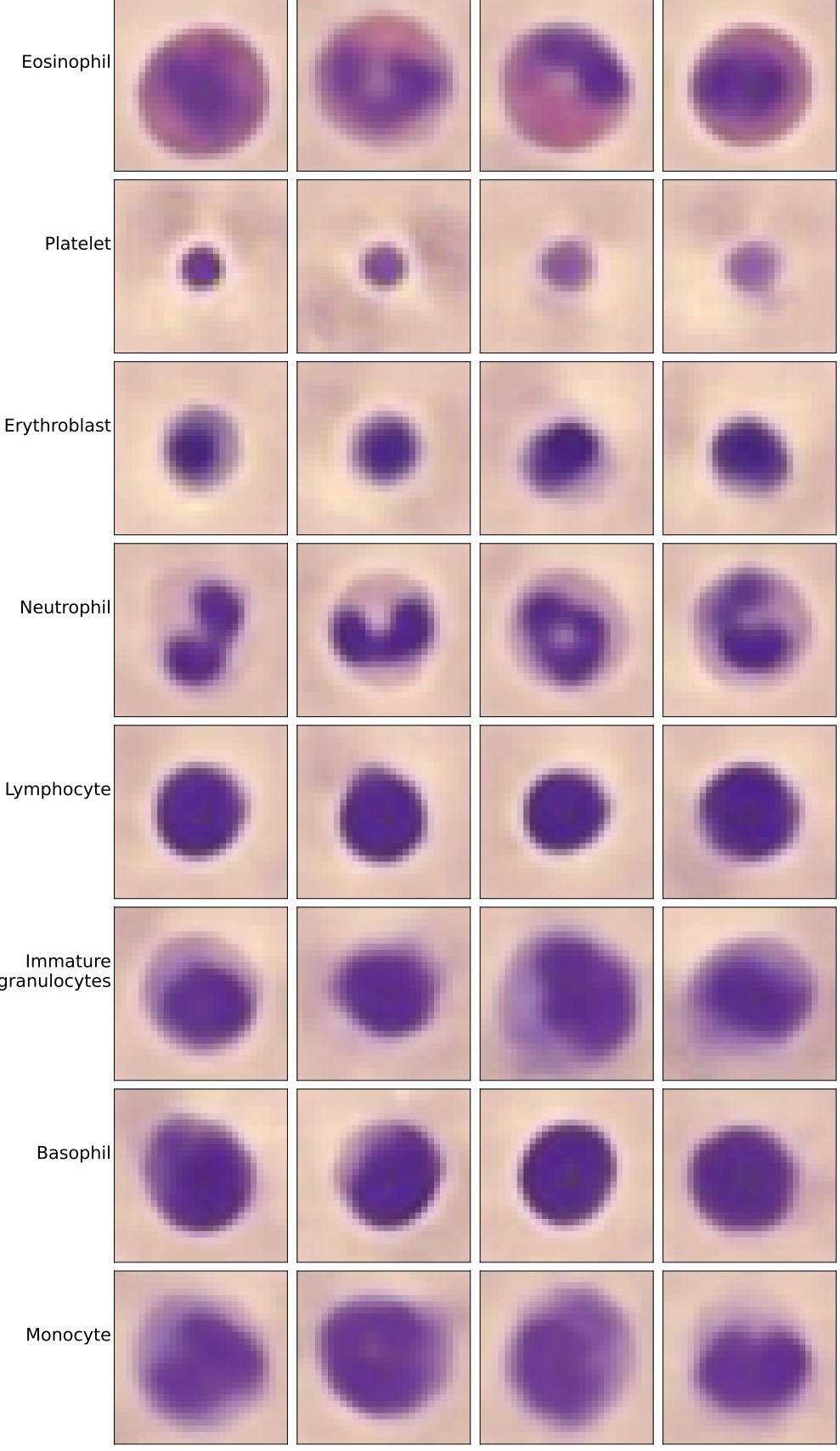

Figure 8: Conditional sampling. Each row corresponds to a different class.

## F   Concept Extraction

In the following we show an example of the concept-traversal plots we exploit to extract interpretable concepts. Latent traversal plots are obtained gradually twisting (increasing or decreasing) a latent dimension while keeping the other elements fixed. These modified representations are reconstructed and the effect of changing a single dimension can be observed. This allows to leverage a human annotator to potentially associate concepts to generative factors by describing how reconstructions change at the varying of the latent. More specifically we traverse the latent space using class medoids (real instance whose encoding was closest to the corresponding latent mean 9) to capture label-relevant concepts.

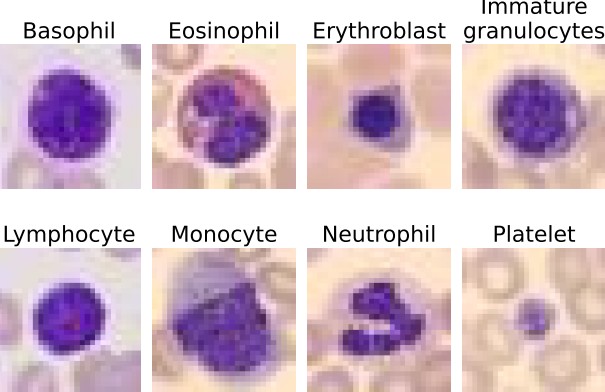

Figure 9: Class medoids

in Figure 10 we present the plot for the medoid of class Erythroblast. It is intuitive that certain dimensions, such as the first, control the darkness of the image, while others, like the third and last, influence the size of the membrane. The shape of the nucleus appears to be modulated by the fourth dimension, and the overall cell size is affected by the eighth and fourteenth dimensions. This reasoning can be extended to all generative factors. Once each dimension is associated with a specific concept, the process is complete, making the concepts ready for explanation.

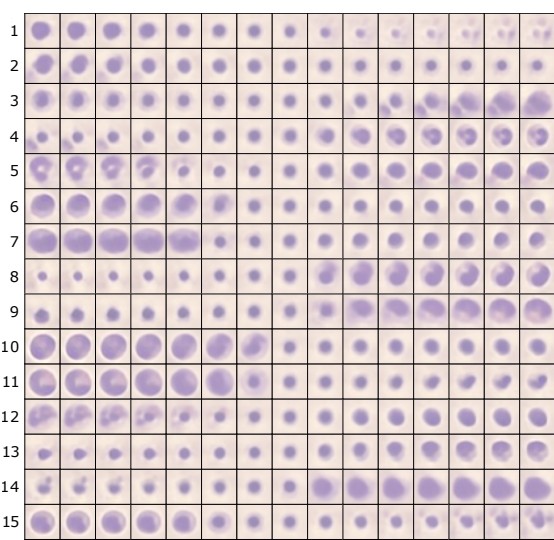

Figure 10: Latent traversal plot of the 15 label-relevant dimensions for Erythroblast.

Furthermore, we provide additional evidence for the meaningfulness of the learned latent space through a small study involving five external human annotators. Specifically, we asked the annotators to examine the rows of Figure 10 and identify the conceptual change they perceived in the images. We report the results using a disagreement matrix among annotators.

|       | $A_1$  | $A_2$  | $A_3$  | $A_4$  | $A_5$  |
|-------|--------|--------|--------|--------|--------|
| $A_1$ | 0.0000 | 0.3333 | 0.2000 | 0.3333 | 0.0667 |
| $A_2$ | 0.3333 | 0.0000 | 0.2667 | 0.2667 | 0.2667 |
| $A_3$ | 0.2000 | 0.2667 | 0.0000 | 0.2667 | 0.1333 |
| $A_4$ | 0.3333 | 0.2667 | 0.2667 | 0.0000 | 0.2667 |
| $A_5$ | 0.0667 | 0.2667 | 0.1333 | 0.2667 | 0.0000 |

Table 6: Disagreement Matrix between Annotators (A). Disagreement quantifies the proportion of elements in the sequences where two annotators assigned different labels. It is computed as the number of differing elements at corresponding positions divided by the total sequence length.

We conclude this brief analysis by noting that the average negative entropy of the assigned labels for each latent dimension was 0.42. Since an entropy of 0.72 would correspond to an agreement among four out of five annotators, this result further supports the meaningfulness of the learned latent space.

## G  Counterfactuals

We provide additional examples of the counterfactuals and concepts generated with our technique for a qualitative analysis in Figure 11. Explanatory images are clear, in-distribution and differences are evident. It is worth mentioning that blurriness in the generated output is due to the compressed latent representation and not to our counterfactual generating technique. This could be of incentive to couple our proposal with more powerful generative models. On the other hand, sharing the label-irrelevant latent dimensions evidently ensures a conceptual similarity as original images and explanations tend to share high level generative factors like inclination or position of the cell in the image. Associated concepts appear clear, pertinent and correctly depict the most relevant changes applied to the input to obtain the explanation. In that regard, the choice of the number of concepts to present is crucial. If the number is too high, certain concepts may capture insignificant variations, reducing the interpretability of the explanations and potentially confusing users.

## H  Experiment

### H.1  Statistics

We expand on the metrics of ACCAM and MIE presented in the experiment results of our contribution. More precisely, ACCAM consists of the accuracy of users on instances for which they do not follow the machine suggestion. Recall from the study design section 6.2.1 that users observe the machine prediction in the `Label` setting and additional explanations in the `Label+Explanation` setting when their first prediction is not the same as the one of the machine. We now call $\hat{y}_i$ the model prediction on $i$-th instance and $\hat{y}_i^H$ the final prediction of human user on $i$-th instance after feedback. We call $n_{diff}$ the number of instances where users final prediction differs from the one of the machine. This is defined as follows:

$$n_{diff} = \sum_{i=1}^{n} \mathbb{1}(\hat{y}_i \neq y_i^H)$$

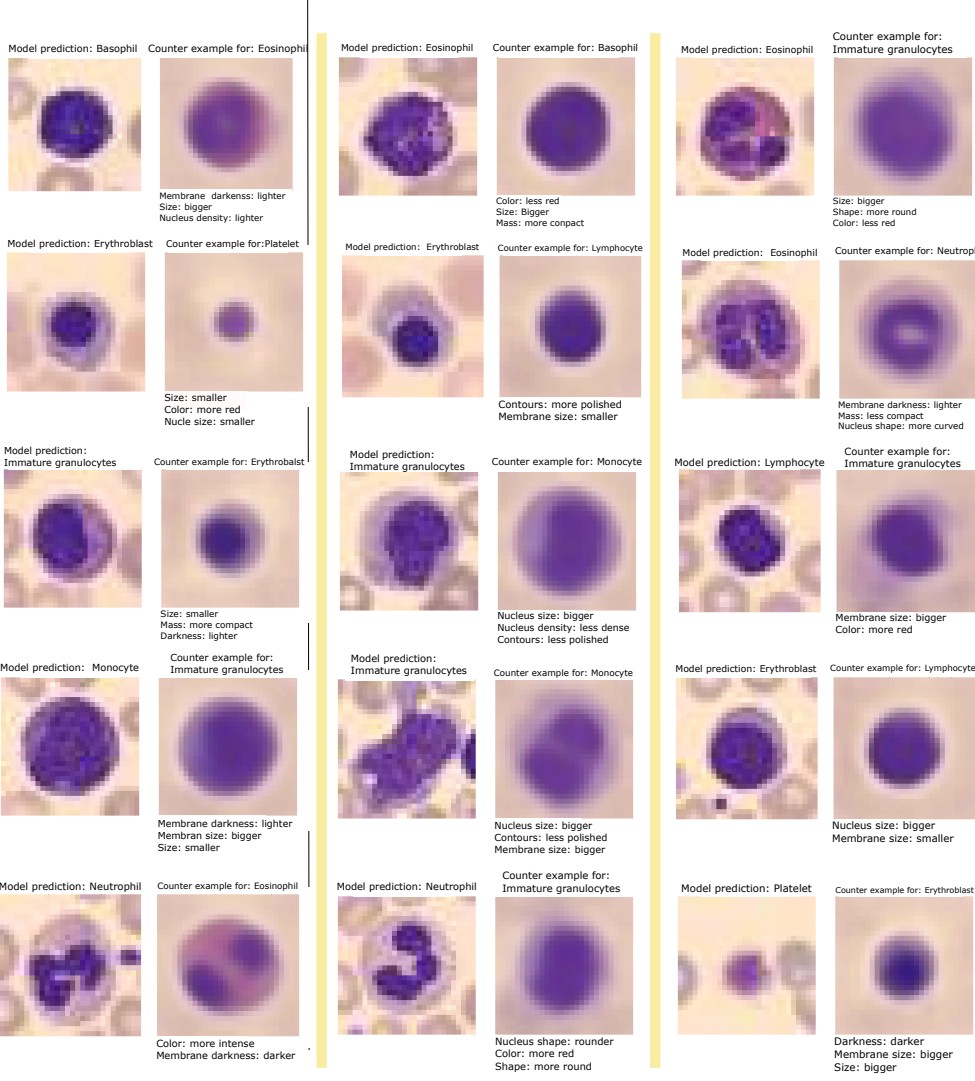

Figure 11: Examples of the generated counterfactuals.

We call $n_{ACCAM}$ the number of instances where users final prediction is correct but this differs from the one of the machine. This is defined as follows:

$$n_{ACCAM} = \sum_{i=1}^{n} \mathbb{1}(\hat{y}_i \neq y_i^H \wedge y_i^H = y_i)$$

where $y_i$ is the true label of $i$-th instance. We can now present ACCAM measure:

$$\text{ACCAM} = \frac{n_{ACCAM}}{n_{diff}} \tag{22}$$

The MIE statistic is instead computed as the number of instances where users were originally correct and wrongly changed their choice after seeing machine feedback. We call the original users predictions on the $i$-th instance before the optional machine feedback as $y_{i,bf}^H$ and we define the number of instances where

users were originally correct as $n_H^+$:

$$n_H^+ = \sum_{i=1}^{n} \mathbb{1}(y_i = y_{i,bf}^H)$$

We now define $n_{MIE}$ as follows:

$$n_{MIE} = \sum_{i=1}^{n} \mathbb{1}(y_i = y_{i,bf}^H \wedge y_i \neq y_i^H)$$

We conclude presenting the MIE statistic:

$$\text{MIE} = \frac{n_{MIE}}{n_H^+} \tag{23}$$

## H.2 Helpfulness of Explanations

From the correlation plots in Figure 4, it appears evident that predictions provided users of an additional help linearly across skill levels. In contrast to this, explanations seem to have the potential to flatten final scores, as the slope of regression line suggests, therefore allowing users across all skill levels to perform well on the task.

Table 7: Density imbalance Scores across skill levels

| Variables | Density imbalance Scores | | | |
|---|---|---|---|---|
| | Q1 (b-l) | Q2 (b-r) | Q3 (u-r) | Q4 (u-l) |
| $\text{ACC}_{bf}$, $\text{ACC}_{af}$ | $-0.073$ | $-0.415$ | $0.224$ | $0.668$ |
| $\text{AGR}_{bf}$, $\text{ACC}_{af}$ | $0.198$ | $-0.277$ | $0.129$ | $0.583$ |

To further investigate this phenomenon, we present in Figure 12 Gaussian density plots of the data points and analyze quadrant-wise density imbalance scores. Specifically, we overlay the data points from the scatter plots in Figure 4 for both versions of our experiment, highlighting regions of space using a Gaussian kernel density estimate to visualize the prevalence of data from either the `Label` or `Label+Explanation` version of the user study. By dividing the plane into four quadrants, we identify regions where: (i) low-skill users receive little help (bottom-left, Q1), (ii) high-skill users receive little help (bottom-right, Q2), (iii) high-skill users receive substantial help (top-right, Q3), and (iv) low-skill users receive substantial help (top-left, Q4).

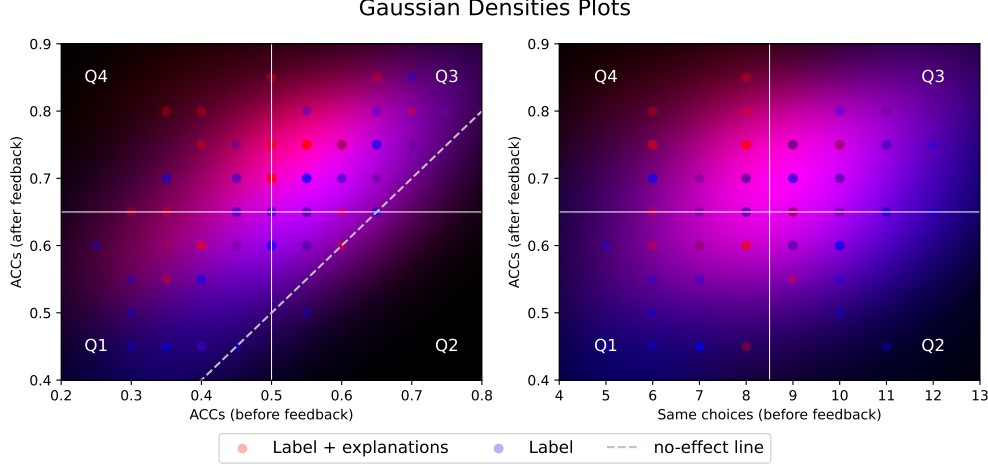

Figure 12: Gaussian densities plots. The coloring depicts the prevalence of points from `Label` experiment or `Label+Explanation` experiment. The latter presents points associated with greater help for less skilled users.

The red predominance in the upper-left quadrant of both plots is evident while bottom-left and upper-right quadrants appear to be equally shared. On the other hand the bottom-right quadrants appears to be mostly blue dominated. This is further supported by the quadrant-wise density imbalance scores of Table 7 where values of the indicator range from 1 to 0 and positive values indicate red dominance while negative values blue dominance. This analysis demonstrates that providing explanations, rather than just model predictions, significantly helped less skilled users achieve competitive performance scores and further validates our proposal.

### H.3 Machine Feedback as a User Training Mechanism

To better understand the impact of explanations on users' ability to complete the task, we analyze the pattern of cumulative errors. Examining cumulative errors helps reveal how mistakes are distributed as the number of interactions with the model increases. In Figure 13, we present the experimental results across all three settings. Notably, in the `None` setting, errors appear to be evenly distributed across questions. In contrast, the `Label` and `Label+Explanation` settings exhibit a distinct pattern, with error rates increasing initially but leveling off significantly after a few interactions with the model. The data reveals that the majority of errors occur within the first 12 questions (nearly half of the experiment), while the last 7 questions account for only 12% of the total mistakes. This strongly indicates the presence of a training effect driven by the interactive framework, especially as the decline in errors occurs immediately after the peak error rate, which coincides with more frequent model interactions.

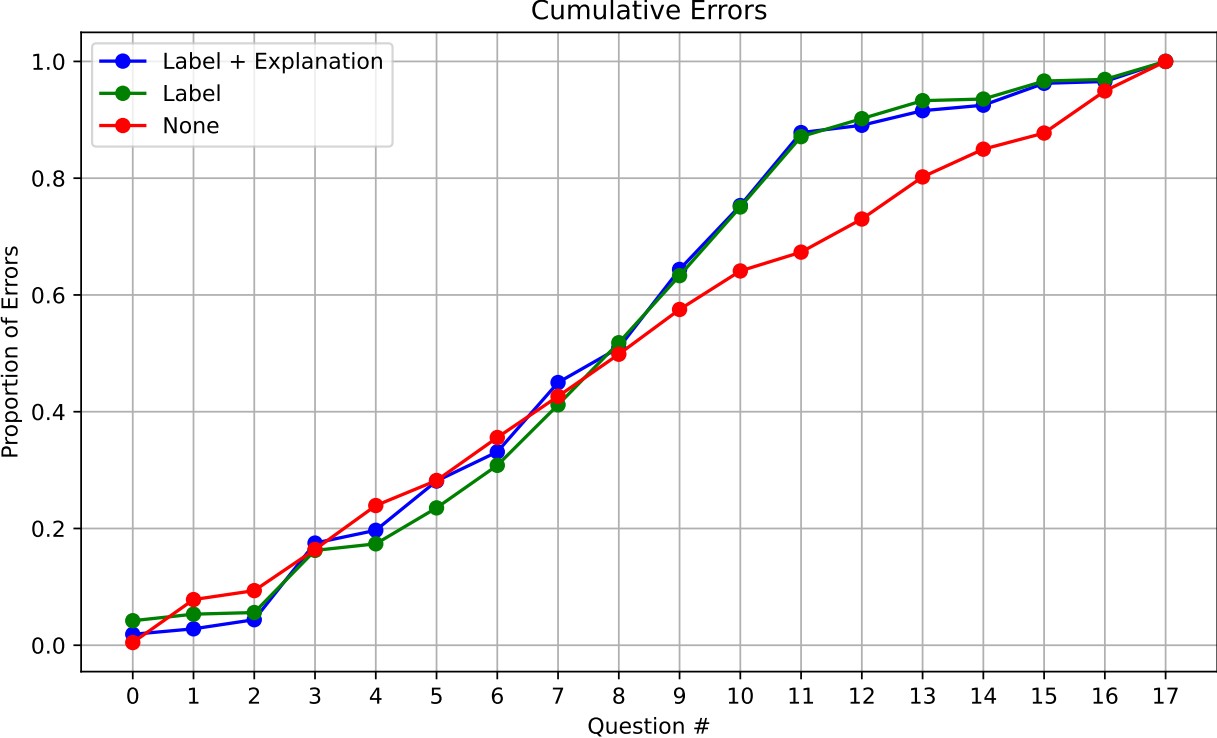

Figure 13: Cumulative proportion of errors made by users across questions. In the `None` setting, errors are evenly distributed across questions, while in the `Label` and `Label+Explanation` settings, users progressively reduce their mistakes, with errors diminishing significantly after sufficient interactions with the model.

### H.4    User Study Preparatory Stage

Given the inherent difficulty of the task users are tackling and given most non-expert users are not familiar with blood cell images, each participant goes through a brief training stage before the beginning of the experiment. In addition, in the Label and Label+Explanation versions of our experiment, users receive an introduction to what the interactive stage consists. For the Label+Explanation version we show this procedure in Figure 14. The training, depicted in Figure 15, consists in showing users images and the corresponding label. More precisely, the first column presents class medoids, while the remaining three columns are populated by random samples from that class. With this, we provide users with a prototypical observation together with information about the variability inherent to each class. In that regard, class medoids consist in the real images whose latent representation was closest to the corresponding latent class mean.

**Training session: interaction**

In the case of disagreement the model will provide a counter example for the user choice and highlight changes in visual features. We briefely mention what these features are and how they behave.

Features refer either to the entire cell, or to one of its three main components:

- **Nucleus** (inner and central part of the cell);
- **Membrane** (lighter and external part);
- **Contours** (borders of the cell).

These can be modified according to their **size, shape, color, darkness, mass** and **density**.

Below is an example of the feedback you would receive in case of disagreement. The original image is shown on the left, together with the prediction by the agent (which differs from yours). On the right is shown a counter-example for the class chosen by you (Eosinophil in this case), i,e., how the image should look like for the agent to agree with your choice, together with the most relevant feature changes applied to produce the counter-example.

Model prediction: Neutrophil          Counter example for: Eosinophil

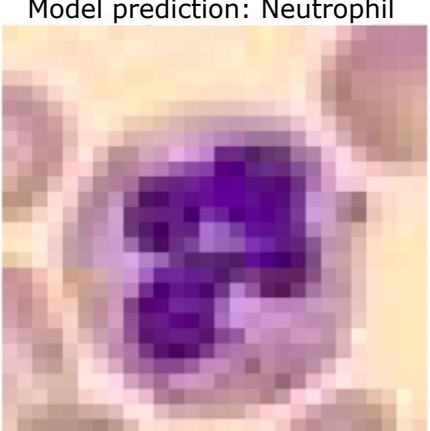 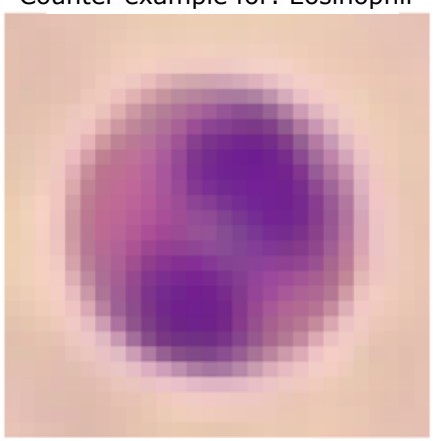

Color: more red
Shape: more round
SIze: bigger

Indietro          Avanti          Cancella modulo

Figure 14: Explanation provided to users of the interactive process

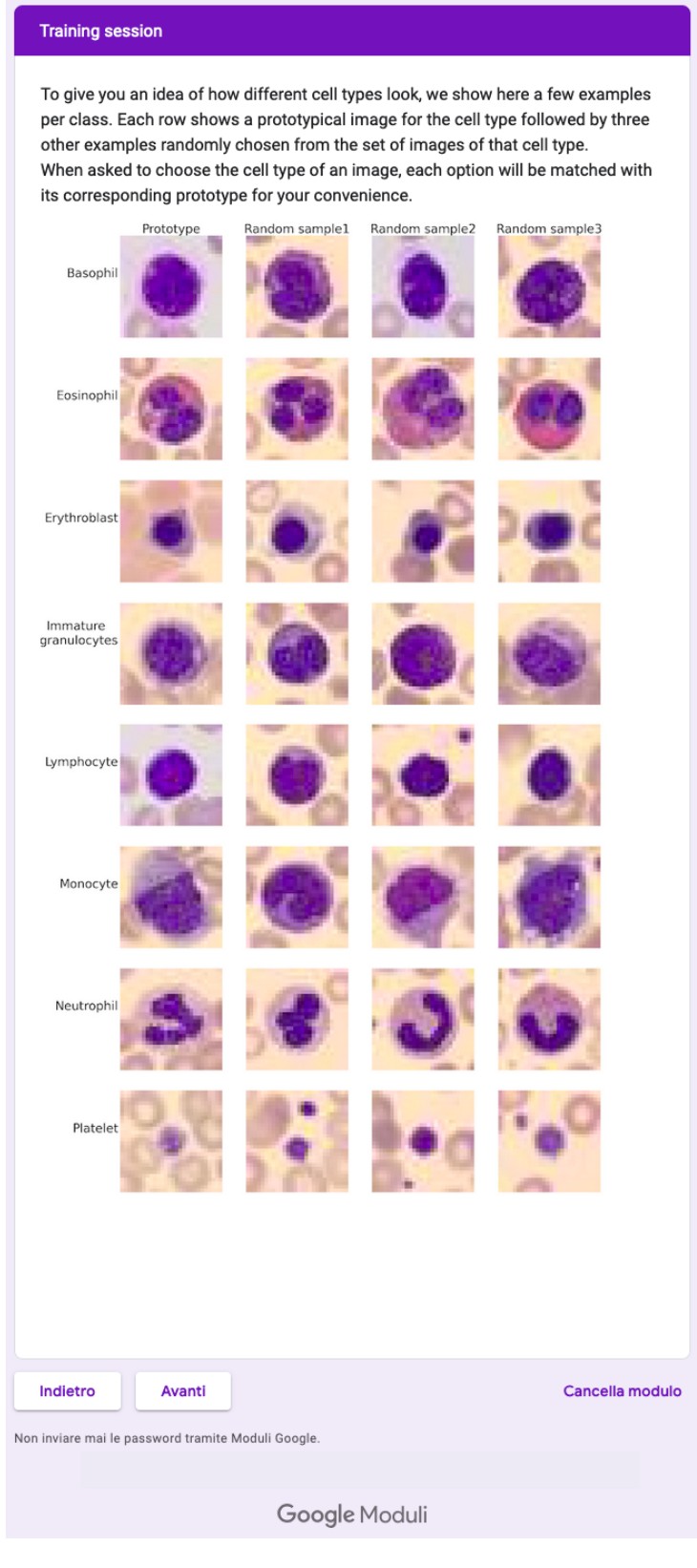

Figure 15: Training session for users.

### H.5 Interface

We present the user interface for the `Label` variant of our experiment and the `Label+Explanation` variant of our experiment. In both variants users are presented a question in the form depicted in Figure16. In case of agreement with the model users jump to the next question after being informed. In the case of disagreement with the model, for the `Label` version, the interface is presented in Figure 17. For the `Label+Explanation` version of the experiment the interface for disagreement is shown in Figure 18.

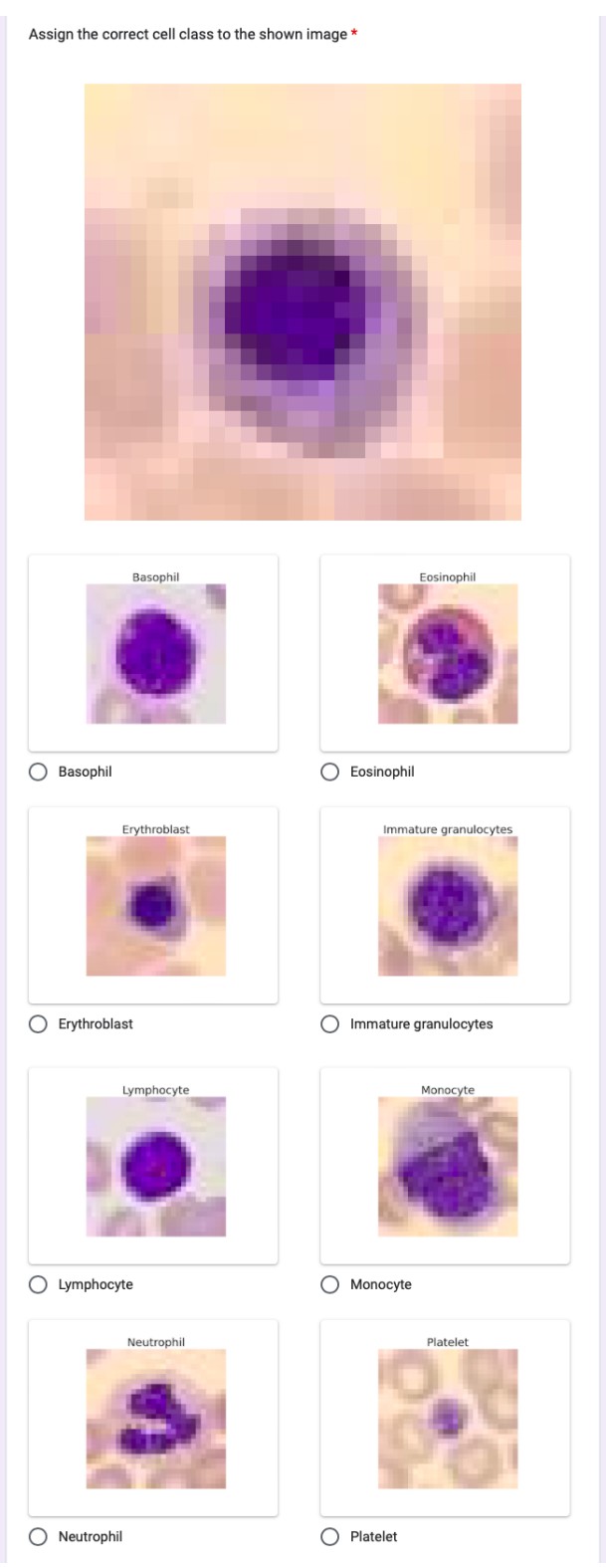

Figure 16: Question example

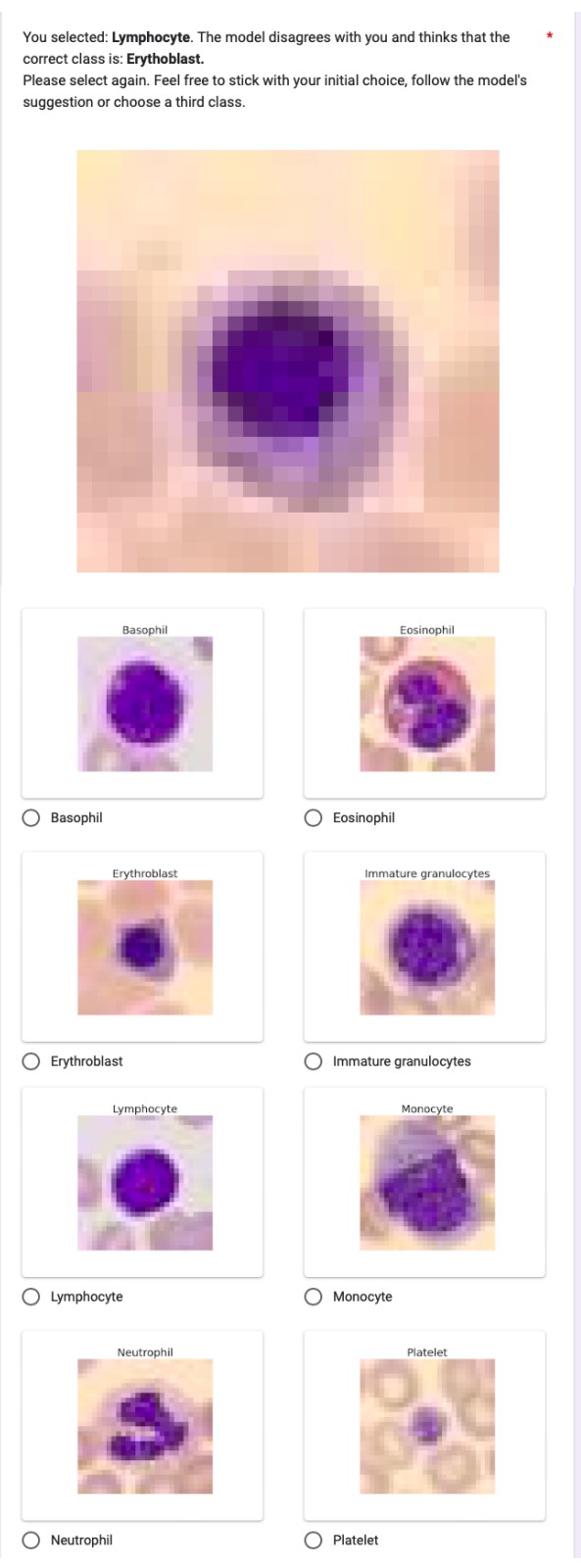

Figure 17: Example of disagreement interface for `Label` version of the experiment

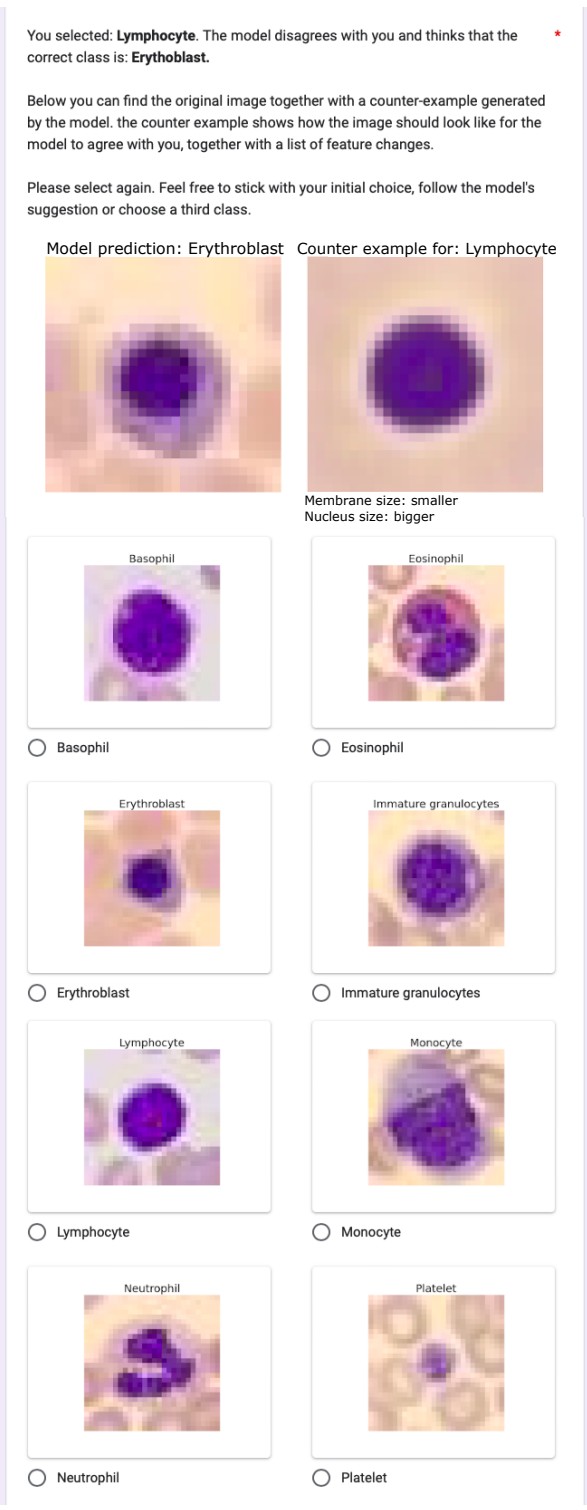

Figure 18: Example of disagreement interface for `Label+Explanation` version of the experiment

