# OpenReview forum: "Interactive Classification with Real-Time Contrastive Explanations"
_TMLR — Rejected by TMLR_

### Review · Reviewer_4uXV · 2025-01-30

**Summary Of Contributions:**

The paper presents the use of counterfactual images to improve performance when a user classifies blood cells.

**Audience:**

No

**Broader Impact Concerns:**

I don't think the results from this paper will have much impact. The counterfactual images that are generated do not appear higher quality than the majority of methods. The user study results don't show a significant trend to conclude an improvement. Also the use of non-experts in the study doesn't align with where I would expect this model to be deployed.

**Claims And Evidence:**

No

**Requested Changes:**

Adjust the claims to reflect existing work.
Adjust the claims to reflect the significance testing results.
Justify why a custom counterfactual generation approach was chosen.

**Strengths And Weaknesses:**

> our proposal is the first interactive classification framework that leverages an interpretable counterfactual generating technique that operates without concept supervision while enabling real time collaboration with users


There has been an interactive user study involving counterfactuals: https://arxiv.org/abs/2304.00487


> 5. Counterfactual Generation

I find the proposed counterfactual generation method overly complicated and also the resulting images don't look as clear as they likely could with other methods that exist. I believe the authors should justify why they used this method and not the others that exist.


> evaluate non-expert user performance in a cell type prediction task

Why not evaluate with experts? What is the use case for a non-expert here? Wouldn't these tools be used by experts? I don't see how this datapoint is useful as the visual explanation would likely help an expert more


> Table 1

It is unclear how these results are significant. The paper states "where explanations helped up to 12% of users outperform the machine." but this appears to be a similar amount to just showing the user the label.

Comparing Label and Label+Explanation using a t-test I don't find the result significant. Here is my computation that yields pvalue=0.097 meaning 9.7% of the time these results are different just based on chance. Perhaps a better test can be used but this is a basic construction.

```
import scipy.stats
scipy.stats.ttest_ind_from_stats(
    mean1=63.99,
    std1=10.45,
    nobs1=20,
    mean2=69.08,
    std2=8.39,
    nobs2=20,
)
```

---

> ### Author Response · Authors · 2025-02-27
> **Answer to reviewer 4uXV**
>
> We would like to thank the reviewer for the insightful and helpful feedback. We applied the requested changes and answer your questions below.
>
> **Related work on user studies with visual counterfactual explanations**
>
> We modified the related work to mention this user study related to explainability for the visual domain (Interactive classification subsection right before ‘but no approach has yet tackled interactive classification for the image domain’. \citep{} does not allow highlighting).
>
> It is worth mentioning that this study does not answer the same questions of our study as users exclusively quantify their confidence in the model choice but do not operate a final classification. Since there is no evaluation of the impact of explanations on the ability of users to solve the task, our claim that our proposal is the first studied framework for the image domain where users obtain real-time feedback to solve a classification task still holds.
>
> **On complexity of the proposal with respect to existing methods**
>
> Our proposed method is not more complex than existing approaches. Many existing techniques leverage auxiliary generative models to explain a model output ([1,2,3,4,5,6,7,8,9]) while we directly integrate this component in our model architecture. As a result, training our model is an efficient and a smooth procedure.
> Our optimization process is performed in two steps. First, latent dimensions are rotated to enable efficient one-dimensional sampling. Second, the sampled expected value is mapped back to the original space by inverting the rotations. This procedure ensures that our approach remains both computationally efficient and simple to implement. Furthermore, as demonstrated in Appendix D.2, our method achieves efficient generation times that scale with model architecture size and significantly outperform existing gradient-based optimization techniques.
> Since our goal is to assess the impact of explanations in an interactive classification setting, it is essential to provide users with real-time explanations. To further clarify this, we have updated the ‘Related Work - Contrastive Explanations’ section of the manuscript, highlighting why our approach is preferable to existing methods.
>
> **Evaluation with experts**
>
> We perform a user study with non-expert users because we are interested in evaluating the effect of explanations in a challenging classification task which users struggle solving on their own.
>
> Interestingly our study findings highlight the great potential of explanations to align performances between users of various skill levels therefore suggesting that level of expertise may not so strongly affect usefulness of explanations. Finally such tools may not be exclusively used by experts but could find utility in human-learning contexts as in Appendix H.2 we show the potential of these methods as a users training mechanism.
>
> **Statistical significance of results**
>
> Results are indeed statistically significant for accuracy and agreement statistics. Only the results of ACCAM are not. Finally, mean difference hypothesis testing on MIE yields non significant differences but this is the desired result.
>
> Here is the relevant code and the results:
>
> ACUCRACY:
> - scipy.stats.ttest_ind_from_stats(
>     mean1=63.99,
>     std1=10.45,
>     nobs1=50,
>     mean2=69.08,
>     std2=8.39,
>     nobs2=50,
> )
> - statistic=-2.6856 ,pvalue=0.0085
>
> AGREEMENT:
> - scipy.stats.ttest_ind_from_stats(
>     mean1=70.8,
>     std1=13.97,
>     nobs1=50,
>     mean2=78.57,
>     std2=13.92,
>     nobs2=50,
> )
> - statistic=-2.7859 ,pvalue=0.0064
>
> ACCAM:
> - scipy.stats.ttest_ind_from_stats(
>     mean1=24.8,
>     std1=21.64,
>     nobs1=50,
>     mean2=29.14,
>     std2=22.2,
>     nobs2=50,
> )
> - statistic=-0.9898= ,pvalue=0.3246
>
> MIE:
> - scipy.stats.ttest_ind_from_stats(
>     mean1=16.24,
>     std1=13.2,
>     nobs1=50,
>     mean2=16.49,
>     std2=13.55,
>     nobs2=50,
> )
> - statistic=-0.0936 ,pvalue=0.9255
>
> Please note that statistics were computed for each single user (total 50 of them) and then results were aggregated computing mean and standard deviation of distributions. Thus,  the number of observations to use in the t-test is 50 (and not 20, which is the value you used in your code snippet).
>
> We adjusted the paper to further clarify how statistics are computed and what results are significant and what are not.
>
> **References**
>
> 1. Cdeepex: Contrastive deep explanations.
>
> 2. Generative counterfactual introspection for explainable deep learning.
>
> 3. Generative causal explanations of black-box classifiers.
>
> 4. Explaingan: Model explanation via decision boundary crossing transformations.
>
> 5. Vae-ce: Visual contrastive explanation using disentangled vaes.
>
> 6. Leveraging latent features for local explanations.
>
> 7. Adversarial counterfactual visual explanations.
>
> 8. Diffusion visual counterfactual explanations.
>
> 9. Latent diffusion counterfactual explanations.

---

### Review · Reviewer_8zG8 · 2025-02-05

**Summary Of Contributions:**

The paper introduces an interactive classification framework that provides real-time contrastive explanations to help users understand and refine model predictions. The authors present a method to generate and update these explanations dynamically, allowing the user to focus on features that distinguish a chosen class from alternatives. In their evaluation, they show that contrastive feedback improves both model performance and user understanding, and discuss how these explanations facilitate more effective collaboration between humans and ML systems.

**Audience:**

Yes

**Broader Impact Concerns:**

The authors provide a useful Broader Impact Statement. I don't have any concerns on the ethical implications of their work beyond that.

**Claims And Evidence:**

Yes

**Requested Changes:**

1. Introducing a rigorous causal model (or, at minimum, clarify the assumptions) to justify the use of terms such as “counterfactual distribution" would drastically improve the paper. To that end, the authors can provide theoretical or empirical validation (or diagnostics) to confirm that the generated counterfactuals plausibly reflect causal interventions.

2. Offering empirical evidence (e.g., ablation studies or sensitivity analyses) showing that the class-conditional latent representations are well modeled by a mixture of Gaussians. If the assumption is too restrictive, consider incorporating a more flexible latent model or discussing its limitations.

3. Including ablation studies or qualitative analyses (for instance, visualizations or case studies) that clearly show how the separation of label-relevant and label-irrelevant dimensions leads to meaningful, interpretable changes in the generated counterfactuals.

4. Besides the user study, adding evaluations using objective metrics (e.g., measures of validity, proximity, and likelihood) and compare these with baselines or established counterfactual methods.

**Strengths And Weaknesses:**

**Strengths**

1. The paper proposes a unified framework that integrates counterfactual explanations directly into an interactive classification setting. The interactive component—where humans can iteratively work with the system distinguishes the work from many static post-hoc explanation methods.

2. By splitting the latent space into label-relevant and label-irrelevant dimensions, the approach aims to gain precise control over the decision boundary. This disentanglement is central to ensuring that counterfactuals reflect meaningful changes, and the paper provides mathematical definitions for candidate counterfactuals (see the conditions P1 and P2 and Proposition 1, where the candidate set S1 and S2 are defined). The idea of “trading off” the likelihood (as measured by the generative model) with the proximity in latent space is an elegant method for balancing quality (likeliness) and similarity (proximity).

3. The paper formalizes the counterfactual search in latent space. For example, Proposition 1 defines the set of candidate counterfactuals in terms of latent segments—from the original point to the class centroid (S1) and along the decision boundary (S2).

4. The includes a user study on a challenging human–machine classification task. The study shows that users collaborating with the system not only improve their performance but in some cases even exceed machine performance.


**Weaknesses**

1. The paper repeatedly uses terms such as “counterfactual distribution” without establishing a rigorous causal framework. There is no explicit causal model or intervention analysis provided, so the claim that the generated counterfactuals reflect a “counterfactual distribution” is questionable. The method does not address whether the assumptions required for causal counterfactuals (for example, the conditions necessary for identifiability in causal inference) are met.

2. The framework is built on the assumption that the class-conditional data distribution in the latent space is well approximated by a mixture of Gaussians. In practice, high-dimensional image data rarely conform to such a simple distribution, and the paper does not provide a discussion or evidence that this assumption holds on realistic datasets. The derivations in Section 5 (e.g., the definition of counterfactual candidates and the subsequent expectation over these candidates) depend heavily on this assumption. The sensitivity of the method to deviations from the Gaussian assumption is not explored.

3. Although the model is designed to extract “interpretable concepts” from latent dimensions, the paper does not provide quantitative or qualitative evidence that these dimensions correspond to semantically meaningful changes in the input space. There is no ablation or detailed analysis showing how much the label disentanglement contributes to generating explanations that users can understand.

4. The primary quantitative evaluation comes from a user study. While this is valuable, there is little discussion of objective metrics (e.g., measures of validity, proximity, or likelihood) that are commonly used to evaluate counterfactual explanations. It remains unclear how the generated counterfactuals compare with those produced by existing methods in a standardized benchmark setting.

---

> ### Author Response · Authors · 2025-02-27
> **Answer to reviewer 8zG8**
>
> We would like to thank the reviewer for the helpful feedback. This was very appreciated. We applied the requested changes to the revised verison of the manuscript which we uploaded. We answer your questions below.
>
> **Counterfactual distribution and causal framework**
>
> We understand the term ‘counterfactual distribution’ might be misleading and we substitute it in the paper with ‘counterfactual label distribution’ and clarify this aspect.
>
> More precisely, our approach models the latent distributions of each class as gaussian distributions. With ‘counterfactual distribution’ we refer to the latent distribution of the class which the returned counterfactual example should belong to.
>
> In practical terms, given an instance x classified as class y_1 and a user query for class y_2 the counterfactual distribution is the gaussian distribution of class y_2 modelled in the latent space by the model.
>
> Although we understand and share the interest for causal frameworks behind counterfactual generation techniques as this allows direct causal interventions, this requires explicit knowledge of casual graphs, which is unavailable for most real-world datasets. As many existing works in the counterfactual explanations domain ([1,2,3,4,5,6,7,8,9,10]) do not leverage underlying causal frameworks to their proposals, we clarified  in the related work that for these very reasons we opted for an approach with broader applicability. Nonetheless, we believe that combining the proposed technique with a causal framework could be a very interesting development for future work.
>
> **Evidence of shaping latent space as a mixture of gaussians**
>
> We understand the doubts posed by the reviewer with regard to latent space distributions as this is a requirement for an effective implementation of our method. More precisely the topic for image classification was first explored by paper [11]. Their approach, which our proposal builds on, effectively reaches the goal of shaping latent spaces as gaussian distributions and it is evaluated on MNIST, CIFAR, IMAGENET and Labeled Face in the WIld (LFW).
> The effectiveness of the approach is also supported by exploiting the densities of such distributions to detect adversarial examples.
> Nonetheless, we added a discussion on the potential limitation of this requirement in the Limitation section of the manuscript.
>
> **Interpretable concepts extraction and lack of an ablation study**
>
> We would like to point out that this detailed qualitative analysis is indeed provided in the Appendix of the paper (Appendix F). There, we present a latent traversal plot to illustrate how variations along individual latent dimensions, while keeping the others fixed, affect the model's generated output. Moreover, the plot provides evidence that these changes can be associated with interpretable concept variations in the model's generations. We acknowledge that these results were not very visible from the main text. We added an explicit mention to the study and its main findings in the ‘concept-based explanations’ section of the main paper.
>
> Furthermore, we added a novel ablation study to evaluate the relevance of the label-irrelevant component of our model architecture. Results highlight the importance of the label-irrelevant encoder in ensuring proximity between original inputs and generated explanations (see Appendix D).
>
> **Lack of quantitative evaluation for counterfactual explanations**
> An extensive quantitative evaluation of the counterfactual explanations produced by our method is provided in Appendix D. There we compare our proposal with the other existing counterfactual generating technique leveraging unsupervised interpretable concepts that we are aware of. We also provide an ablation study for the counterfactual generation process, by comparing our optimization with a simple interpolation method. We compute FID, COUT and S3 scores to capture likeness, validity (or effectiveness in targeting validity) and proximity of the generated explanations.
>
> These quantitative results are briefly mentioned at the end of the introduction. To better highlight them, we added a paragraph describing the evaluation and its main findings in the experimental section of the main paper.
>
> ** References**
>
> 1. Cdeepex: Contrastive deep explanations.
> 2. Generative counterfactual introspection for explainable deep learning.
> 3. Generative causal explanations of black-box classifiers.
> 4. Explaingan: Model explanation via decision boundary crossing transformations.
> 5. Vae-ce: Visual contrastive explanation using disentangled vaes.
> 6. Leveraging latent features for local explanations.
> 7. Adversarial counterfactual visual explanations.
> 8. Diffusion visual counterfactual explanations.
> 9. Latent diffusion counterfactual explanations.
> 10. Counterfactual explanations without opening the black box: Automated decisions and the GDPR
> 11. Rethinking feature distribution for loss functions in image classification

---

### Review · Reviewer_axKM · 2025-03-26

**Summary Of Contributions:**

The paper presents a framework for real-time generation of counterfactual explanations for image classifiers. The approach is based on a combined autoencoder and Gaussian mixture classifier. The authors manually identify the latent dimensions of the autoencoder with textual concept descriptions and produce counterfactual images based on a heuristic averaging of latent embeddings along a potentially relevant trajectory in latent space. A user study presents some hints that the method may indeed be helpful to unskilled users.

Overall, it is a nice idea that could be a step in the right direction. However, the execution falls short on several fronts, unfortunately limiting the potential impact of the work substantially.

**Audience:**

Yes

**Broader Impact Concerns:**

Human data: You state that the "study was conducted in compliance with the TMLR Code of Ethics." However, the TMLR Code of Ethics, point 4, asks if the research might "Contain human subject experimentation and whether it has been reviewed and approved by a relevant oversight board." As your study contains human data, please state whether it has been approved by an oversight board.

**Claims And Evidence:**

No

**Requested Changes:**

Claims not supported by evidence need to be rephrased or removed. This pertains particularly to the specific instances listed in the Weaknesses section above. Also, if you want to maintain the language around disentangled latents, you need to provide empirical evidence for it. Finally, the conclusions from the user study need to be reduced massively or run a refined study with careful experimental design and statistical analysis framework.

Those parts of the quantitative evaluation (Section 6.1) that are currently in Appendix D should be included in the main manuscript.

**Strengths And Weaknesses:**

### Strengths

 1. Interesting idea addressing an important question
 1. Attempt to perform human validation on a challenging classification task


### Weaknesses

 1. Frequent overly strong claims given the limited evidence
 1. The authors claim to achieve disentanglement of latents, but present no evidence
 1. User study not designed carefully and conclusions not fully supported by rigorous statistical analysis
 1. Method and approach not entirely clear from the main manuscript / details missing
 1. Frequently very "mathy" (unnecessarily so?) but not clear if grounded in first principles



## Detailed comments


### Claims not supported by evidence

The paper is full of strong claims that are not actually supported by evidence, which is very frustrating for readers. I urge the authors to carefully go over the entire manuscript and carefully check each claim whether it is supported by evidence or can be rephrased on a more accurate and nuanced way. Below I list a few examples from the prominent sections, but this list is by no means complete and the other sections also need to be carefully checked.

#### Abstract
- The abstract reads like a revolution, solving all long-standing challenges in counterfactual generation but then the paper's content are disappointing given such strong claims. A few examples below.
- "generating high-quality instances" – The likelihood of generated instances is certainly a weakness also in the current method. They don't look realistic at all. The authors present no evidence that their explanations have high likelihood when using a strong generative model to assess it. The likelihood may be high under their own model, but it's not a good one in any respect. It certainly does not produce realistic examples.
- "promoting label disentanglement" – I could not find evidence that such disentanglement is actually achieved.
- "emphasizing the importance of contrastive explanations" – There is no evidence that contrastive explanations are important.
- In terms of classification performance the method is quite far from state-of-the-art

#### Introduction
- "The study results clearly demonstrate the potential of our approach in enhancing human performance, with some users even surpassing machine performance." – There is no evidence that some users actually surpass machine performance. The higher values could simply be measurement noise that is expected from the 20 samples tested. Such conclusion would need rigorous statistical testing (and, presumably, more data from some individuals to have sufficient statistical power at the level of the individual).

#### Main text

- Section 6.2: Several bold sounding claims, for which it's not clear what they're based on.
	- "Gaussian classification ensures 100% validity on generated explanations" --> Why?
	- "In addition, we facilitate the interaction step by eliminating the need for hyper-parameter configuration, thereby reducing potential confusion for non-expert users." --> Not clear what eliminates the hyper-parameter tuning. The argument becomes clearer if one has read Appendix D and seen Table 2 but you can't assume people read that at this point if it's not in the main paper. Also, I don't buy the argument. Your averaging is one specific hyperparameter setting chosen for the BloodMNIST dataset and you haven't verified that it generalizes to other datasets.


#### Conclusion
- "The results highlighted that explanations are beneficial for users of all skill levels" --> Not the case. No evidence that skilled users benefit.
- "the study underscored the essential role of contrastive explanations in enhancing user understanding and trust, showcasing their pivotal contribution to achieving clear and actionable insights." --> Not the case. No evidence that alternative explanation forms to contrastive are not effective in enhancing user understanding and trust, as no such alternative explanations were included in the user study. Also, no evidence that "clear and actionable" insights were generated by explanations.



### Disentangled latents

The approach rests on the assumption that individual dimensions in the autoencoder's latent space are meaningful. However, this is not expected given the training objective. As you state correctly, "constructing disentangled embedding spaces is not always feasible, as the problem is inherently unidentifiable without additional assumptions or supervision." The latent space can be rotated arbitrarily in the Gaussian model. Hence, there is nothing in your approach (that I can see) that would lead to disentangled latents. It could be that due to unknown reasons it is, but such conclusion would require more empirical evidence than a single latent traversal plot (Fig. 10; that doesn't even look convincing IMHO). Such evidence could be, for instance:

- Show that two (slightly) different models with different initial conditions generate similar latent axes that can be mapped onto each other.

- How do the text explanations below the counterfactual images come about? If they were consistent across annotators and models, this would provide evidence for the latent space to be meaningful.

- Related to the previous point: I may have missed something but the description in section 5.4 only hints at a "human annotator" identifying "meaningful changes". In my opinion, this procedure needs much more scrutiny and detail. A few examples:
	- How was this human annotator chosen? Expert, novice?
	- How many annotators? Did you check for consistency of their textual descriptions of what the latents mean? If so, how and how consistent were they across annotators? If your claim is that the latents are meaningful, humans should identify them consistently.


### User study

The user study is an applaudable effort, but unfortunately does not appear to be designed carefully enough to support the claims you're trying to make. Several issues:

- The study does not attempt to disentangle the contributions of (a) showing a counterfactual image and (b) providing text explanations as both a presented only together. This is an unfortunate design limitation of the study.
- Control conditions are missing: Could the better user results under Label+Explanation be attributed to the two potential confounds that with (a) with an image or (b) text present, the users see more examples and investigate/compare them more carefully? It is not strictly correct to conclude from the given study that the explanations are useful. To draw such conclusion, one would have to show "placebo" conditions where random examples from the model-predicted class are shown and random text explanations are shown. Such controls would account for the potentially increased attention that users pay to the images rather than the explanations per se.
- Table 1 needs statistical hypothesis tests to check which differences are significant. For the claim to hold that explanations are useful, you would have to show that the performance of Label+Explanation is significantly better than Label only. The manuscript reports this difference as significant (p = 0.004). But that’s a post-hoc test. How were multiple comparisons accounted for?
- I did not fully understand what quantities ACCAM and MIE exactly measure. Formulas and/or an illustration via a Venn diagram could help.
- Where is the evidence that RG2 is answered positively? If it's meant to be the higher ACCAM for Label+Explanation than Label, where are the statistics on it? The standard deviations are very large and it would constitute another test which would have to be accounted for in a multiple comparisons procedure.
- The negative answer to RQ3 is an overstatement! The question you posed was whether explanations "can" be harmful or mislead users. Absence of evidence is not evidence for absence! You definite cannot answer this question with "No" given your evidence. This needs to be phrased much more carefully.
- You conclude that you "can confidently assert that the explanations provided are beneficial across all user skill levels". I disagree with this assertion! You show that on average they benefit, but Fig. 4 suggests that, in fact, skilled users with high accuracy before feedback may not benefit (at least from a qualitative glance at the plot). Statistics establishing a benefit for skilled users are lacking.



### Mathiness

The paper contains lots of equations and formal definitions for things that strike me as pure heuristics and could be explained in much simpler terms. If they are grounded in first-principles probabilistic reasoning, this needs to be much more clearly explained. Alternatively (and that's my suspicion), if it's mostly heuristic, the readers would benefit from a more intuitive exposition supported by illustrations and clear motivations rather than complicated equations that are not fully motivated. Examples include but are not limited to:

 - Proposition 1 / Eqs. 7–9: Why the expected value over the two line segments? What motivates it? Wouldn't the point closest to the decision boundary be the best? Or show multiple examples along the trajectory instead of an expectation where it's unclear how close it is to the decision boundary?

 - Definition 1: Reference Fig. 2 (left) directly around Definition 1 in section 5.1. I had to wade through the math and make drawings to figure out what's going on only to find out that a figure exists that illustrates it, but it was two pages later and referenced only at the end of section 5.2.

- Definition 2 / Eq. 11: What exactly motivates this? It seems clear what you do, but I couldn't get a sense for why exactly this procedure. Is it grounded in a principled probabilistic reasoning or some form of heuristic?

---

> ### Author Response · Authors · 2025-04-01
> **Answer to reviewer axKM**
>
> **Quality of explanations** We changed ‘high-quality instances’ clarifying that explanations are very likely according to the learned latent distributions.
>
> **Label Disentanglement**
> We use the term as in [1], distinguishing label-relevant from label-irrelevant dimensions. To clarify this, we updated the text, especially regarding the label classification task. Please note, the model achieves 91% accuracy on Blood-MNIST, slightly below the 96% benchmark [6], but it was designed to be imperfect to study the interactive system behavior with errors.
>
> **Contrastive Explanations** Contrastive explanations are important because they improve interpretability with respect to other explanations [2,3,4] and are user-preferred [5]. We highlight their role in an interactive classification task. We clarified that our focus is on explanations in interactive settings, not specific types.
>
> **Surpassing Machine Performance** We acknowledge that this difference would require larger sample size to extract significant conclusions. We rephrased to be more cautious.
>
> **Validity** We define candidates in the latent space based on two properties, including validity. In Appendix B.1, we prove that under linear decision boundaries, candidates lie on two segments ($S_1$ and $S_2$ in Figure 2). Since Gaussian classification leads to linear boundaries, it ensures validity by design. We updated the text with a reminder to the properties of candidates and clarify this connection.
>
> **Hyper-parameter Tuning** Hyper-parameter tuning is now addressed in the main text. Unlike fixing a hyper-parameter value, our approach provides an optimal solution to the trade-off between likelihood and proximity, ensuring with statistical guarantees in-distribution explanations across domains. Additionally, tuning likelihood is unintuitive for non-experts and can require extensive tuning. Avoiding this, our method is more reliable and user-friendly.
>
> **Improvements and Actionability** In Figure 4, label+explanations case, all points lie on or above $y=x$ (now added), confirming that explanations improve users accuracy, even for experts. See Appendix H.2 for additional statistics. Also, evidence for actionable insights comes from accuracy improvements after feedback, so we added the corresponding p-values to the main text.
>
> **Latent Disentanglement** Following the reviewer's suggestion, 5 annotators labeled each latent axis assigning concepts. Appendix F reports the disagreement matrix and annotation entropy, showing high agreement with minor variation due to domain complexity.
>
> **Study Variants** Examples from the model-predicted class are already present as the class medoids shown beside each class, even in the baseline setting with no counterfactual. While many alternative settings could be conceived, including counterfactual image alone and text feedback alone, experimenting with them would complicate the design and increase the amount of participants (and costs) required to draw conclusions. We believe this would suit better a large experimental study on types of feedback, than a methodological work evaluating an interactive approach to classification.
>
> **Statistics** We mention significance of results in the main text of our manuscript and add p-values to the table. Bonferroni correction changes the level of alpha from 0.05 to 0.0125 and does not affect the significance of results. We also updated the manuscript to present formulas to compute MIE and ACCAM in Appendix H and refer to this change in the main text.
>
> **Overclaiming** We changed the text for our RQ2 claims to better reflect the experimental results. We also revised the definition of RQ3 for clearer goals.
>
> **Expectations along segments** While the closest point to the decision boundary maximizes proximity, counterfactual generation balances likelihood and proximity. We achieve this via expected value computation to prevent out-of-distribution explanations. Note that this is a point of equilibrium as the weights of instances with more proximity and the weights of more likely instances balance out. Finally, instead of generating a trajectory of explanations we return a single instance to minimize the cognitive burden for users throughout the study.
>
> **Figure position** We fixed the inconvenient plot location by moving it closer to formulas and mentioning it earlier in the paper.
>
> **Concept Extraction** We weight squared differences by likelihoods to prevent out-of-distribution components from distorting concept retrieval and reducing explanation clarity. We clarified this in the text.
>
> **Human Data** We assessed the risk of our study using a survey designed by our institution. This confirmed that IRB approval was not necessary.
>
> **References**
>
> 1: Disentangling latent space for vae by label relevant/irrelevant dimensions.
>
> 2: Cdeepex
>
> 3: Generative causal explanations of black-box classifiers
>
> 4: Explaingan
>
> 5: Leveraging latent features for local explanations
>
> 6: https://medmnist.com/

---

### Author Response · Authors · 2025-04-01
**Official Comment for Reviewers**

We would like to thank all reviewers for the insightful feedback. We appreciated the constructive comments that helped us a lot in improving our work. We applied the requested changes to the manuscript and look forward to engaging further in this discussion on the topic of Interactive Classification.

---

### Author Response · Authors · 2025-05-09
**Manuscript status**

Dear editor and reviewers, we would like to know if there is any news about the status of our manuscript, and if thre is any further clarification that is needed on our side.

---

### Decision · Action_Editor_RL7w · 2025-06-10

**Recommendation:** Reject

**Audience:**

Yes

**Audience Explanation:**

The findings, if supported by carefully designed user study and rigorous statistical analysis, would be of interest to the XAI researchers.

**Claims And Evidence:**

No

**Claims Explanation:**

This paper proposes an interactive classification framework that provides real-time contrastive explanations to help users understand and refine model predictions. Based on thorough reviews by expert reviewers and additional feedback from the reviewers after receiving the revision, the paper in its current form appears to still contain many claims that are not supported by convincing and clear evidence. The major concerns specifically include:

- **The user study is not carefully designed**. For example, counterfactual images and text explanations are presented together, making it impossible to disentangle the contributions of either showing a counterfactual image or providing text explanations alone. Control conditions are also missing in the user study, rendering the claims inconclusive. That is, to conclude that the provided explanations are useful, one would have to consider scenarios in which random examples from the model-predicted class as well as random text explanations are shown to the users. Moreover, the paper presents results assisting “non-expert” users and provides no strong justification for this to reflect the real usage of this method (e.g., benefits for skilled users are lacking).

- **Lack of rigorous statistical analysis and significant results**. It remains unclear whether the results in Table 1 are significant. Qualitatively, the generated images of the proposed method do not appear to be of higher quality than existing methods.

- **Interpretation of disentangled latents**. The paper claims that the disentangled latents provide meaningful interpretation. However, this claim is not fully supported at least by strong empirical evidence, given that the problem is inherently unidentifiable. There is no quantitative or qualitative evidence to support that the latent dimensions correspond to semantically meaningful changes in the input space.

After considering the revision, the reviewers and the action editor feel that the revision only contains cosmetic changes to the manuscript, and thereby doesn’t fully address the aforementioned major concerns raised by the reviewers.

**Resubmission Of Major Revision:**

The authors may consider submitting a major revision at a later time.